# Control of seed dormancy and germination by DOG1-AHG1 PP2C phosphatase complex via binding to heme

Noriyuki Nishimura[1,8], Wataru Tsuchiya[2], James J. Moresco[3], Yuki Hayashi[4], Kouji Satoh[1], Nahomi Kaiwa[1], Tomoko Irisa[1], Toshinori Kinoshita[4,5], Julian I. Schroeder[6], John R. YatesIII[3], Takashi Hirayama[7] & Toshimasa Yamazaki[2]

Abscisic acid (ABA) regulates abiotic stress and developmental responses including regulation of seed dormancy to prevent seeds from germinating under unfavorable environmental conditions. *ABA HYPERSENSITIVE GERMINATION1* (*AHG1*) encoding a type 2C protein phosphatase (PP2C) is a central negative regulator of ABA response in germination; however, the molecular function and regulation of AHG1 remain elusive. Here we report that AHG1 interacts with DELAY OF GERMINATION1 (DOG1), which is a pivotal positive regulator in seed dormancy. DOG1 acts upstream of AHG1 and impairs the PP2C activity of AHG1 in vitro. Furthermore, DOG1 has the ability to bind heme. Binding of DOG1 to AHG1 and heme are independent processes, but both are essential for DOG1 function in vivo. Our study demonstrates that AHG1 and DOG1 constitute an important regulatory system for seed dormancy and germination by integrating multiple environmental signals, in parallel with the PYL/RCAR ABA receptor-mediated regulatory system.

[1] Radiation Breeding Division, Institute of Crop Science, National Agriculture and Food Research Organization, 2425 Kamimurata, Hitachiohmiya, Ibaraki 319-2293, Japan. [2] Structural Biology Team, Advanced Analysis Center, National Agriculture and Food Research Organization, Tsukuba, Ibaraki 305-8602, Japan. [3] Department of Molecular Medicine, The Scripps Research Institute, 10550 North Torrey Pines Road, La Jolla, CA 92037, USA. [4] Division of Biological Science, Graduate School of Science, Nagoya University, Nagoya 464-8602, Japan. [5] Institute of Transformative Bio-Molecules (WPI-ITbM), Nagoya University, Chikusa, Nagoya 464-8602, Japan. [6] Division of Biological Sciences, Cell and Developmental Biology Section, University of California, San Diego, 9500 Gilman Drive, La Jolla, CA 92093-0116, USA. [7] Institute of Plant Science and Resources, Okayama University, 2-20-1 Chuo, Kurashiki, Okayama 710-0046, Japan. [8] Present address: Division of Basic Research, Institute of Crop Science, National Agriculture and Food Research Organization, 2-1-2 Kannondai, Tsukuba, Ibaraki 305-8518, Japan. These authors contributed equally: Noriyuki Nishimura, Wataru Tsuchiya. Correspondence and requests for materials should be addressed to N.N. (email: nonishi@affrc.go.jp)

As sessile organisms, plants have evolved a number of unique mechanisms to adapt to environmental changes. Seed dormancy, which is increased during seed maturation, is one strategy in plants to prevent the seeds from germinating under unfavorable environmental conditions[1–4]. The ability to regulate seed dormancy is considered an important trait for the domestication of crops, because reducing seed dormancy leads to pre-harvest sprouting during crop production, while an increase in seed dormancy makes it difficult for the plants to germinate in the field. Thus, seed dormancy and germination are strictly connected by developmental and environmental conditions. Under favorable conditions, the phase transition from seed dormancy to germination is controlled by the plant hormones abscisic acid (ABA) and gibberellin (GA). In seeds, GA is known to induce germination and inhibit seed dormancy, while ABA antagonizes GA signaling[1,4,5]. ABA regulates abiotic stress and developmental responses including seed maturation, regulation of seed dormancy and germination, growth regulation, and stomatal closure[6,7], and has recently been shown to transiently elevate heme levels[8,9]. Heme, which is an iron-binding protoporphrin IX, is a key molecule that regulates diverse biological activities including light respiration, secondary metabolism, and signal transduction[10,11], however, its role in ABA signaling is mostly unknown.

Genetic analyses have identified many loci involved in seed dormancy and germination[1–3,7]. DELAY OF GERMINATION1 (DOG1) encodes a protein of unknown biochemical function and was first identified in Arabidopsis as a major quantitative trait locus (QTL) for an increase in seed dormancy[12]. The degree of seed dormancy was determined by the abundance of DOG1 protein in freshly harvested seeds, therefore, it was proposed that DOG1 is a timer for release from dormancy[13]. While DOG1 has been proposed to function independent of ABA signaling[13], a recent study has shown that both DOG1 and ABA signaling function in both seed dormancy and seed maturation[14]. Thus DOG1 is a pivotal regulator of seed dormancy.

The core ABA signaling mechanism is widely thought to be composed of three major components: Pyrabactin Resistance 1 (PYR1)/PYR1-Like (PYL)/Regulatory Components of ABA Receptor (RCAR) ABA receptors, group A type 2C protein phosphatases (PP2Cs), and subclass III sucrose nonfermenting-1-related protein kinase2s (SnRK2s)[6,7,15–18]. In the presence of ABA, ABA-bound PYR/PYL/RCARs activates SnRK2s through the inhibition of phosphatase activity of PP2Cs. Most group A PP2Cs are negatively regulated by PYL/RCAR ABA receptors and control the activation of target proteins such as subclass III SnRK2 members including SnRK2.2, SnRK2.3 and SnRK2.6/OST1 in an ABA-dependent manner to evoke physiological responses[15,16,19,20]. The snrk2.2snrk2.3snrk2.6 triple loss-of-function mutant was shown to exhibit vivipary and is almost completely unresponsive to ABA, supporting the evidence that these kinases are important for the regulation of seed dormancy and germination regulated by ABA signaling[21–24]. Activated SnRK2s phosphorylate downstream targets including b-ZIP type transcriptional factors ABA INSENSITIVE5 (ABI5) and abscisic acid responsive elements (AREB)-binding factors (ABFs)[7,21,22,25], and the ion channel SLOW ANION CHANNEL-ASSOCIATED1 (SLAC1)[26–28] that is involved in stomatal response[29,30].

The Arabidopsis genome, which includes more than 76 PP2C genes[31,32], has nine group A PP2Cs that function as central negative regulators of ABA signaling[7]. Based on the sequence alignment, the group A PP2Cs can be classified into two subfamilies named ABI1 and ABA HYPERSENSITIVE GERMINATION1 (AHG1). The ABI1 subfamily is formed by ABI1, ABI2, HYPERSENSITIVE TO ABA1 (HAB1) and HAB2, while AHG1 subfamily is formed by AHG1, AHG3/PROTEIN

PHOSPHATASE 2CA (PP2CA), HIGHLY ABA-INDUCED PP2C GENE1 (HAI1), HAI2/AKT1 INTERACTING PP2C (AIP1), and HAI3. AHG1 and AHG3 have been identified as genetic loci that are involved in the ABA response in seed germination[33]. ahg1 and ahg3 mutants show a strong ABA hypersensitive phenotype in germination compared to other PP2C single mutants[34–36]. Since the expression levels of AHG1 and AHG3 are higher in dry seeds and increased during the seed maturation stage or in the presence of ABA, we proposed that the expression levels of PP2C genes in seeds are likely to be a major factor contributing to ABA response in seed germination. A detailed analysis indicated that AHG1 and AHG3 have both overlapping and distinct functions[36]. Consistent with this observation, PYL/RCARs terminated the PP2C activity of AHG3, but not that of AHG1 in the presence of ABA[20], suggesting that AHG1 functions in a unique regulatory system independent of PYL/RCAR ABA receptors.

In order to understand the molecular function and regulation of AHG1 in the ABA signaling pathway, we have conducted experiments to co-purify AHG1-interacting proteins in Arabidopsis using affinity column-based purification. Interestingly, DOG1 has been identified as an in vivo interactor of AHG1. Our epistatic analysis demonstrates that DOG1 acts upstream of AHG1 and reduces the PP2C activity of AHG1 in vitro. Furthermore, we find that DOG1 is an α-helical protein that has the ability to bind both AHG1 and heme. Binding of DOG1 to AHG1 and heme are independent processes, but both are essential for DOG1's function in vivo. Our study unveils a novel regulatory system of seed dormancy and germination regulated by ABA signaling through a DOG1–AHG1 interaction, in parallel with PYL/RCAR ABA receptor-dependent regulation.

## Results

**Physical interaction between AHG1 and PYR1.** We previously identified PYL/RCAR ABA receptors as in vivo ABI1-interacting proteins by a combination affinity column purification, using YFP-ABI1 overexpressing plants (YFP-ABI1ox) with LC-MS/MS[37]. To assess whether PYR1 was able to interact with all nine group A PP2Cs including AHG1, which we previously identified as a central negative regulator of ABA signaling in seeds (Supplementary Fig. 1a)[36], we performed yeast two-hybrid assays (Fig. 1a). A previous study found that ABI1 subfamily members interacted with PYR1 in an ABA-dependent manner[16]. Interestingly, in our investigation, AHG1 subfamily members, except for AHG3, did not interact with PYR1, even in the presence of ABA (Fig. 1a). When HA-PYR1 was co-expressed with either YFP-AHG1 or YFP-AHG3 in Nicotiana benthamiana, HA-PYR1 co-immunoprecipitated with YFP-AHG3 in an ABA-dependent manner, but not detectably with YFP-AHG1 (Fig. 1b). Some PYL/RCAR-GFP fusion proteins, which interact with ABI1 subfamily members in vivo, have been reported to localize in both the cytoplasm and the nucleus[15,16,38]. With the exception of HAI2/AIP1, YFP fused to AHG1 subfamily members predominantly localized in the nucleus when transiently expressed in N. benthamiana protoplasts, whereas YFP fused to all the ABI1 subfamily members and HAI2/AIP1 were observed in the cytoplasm and the nucleus, consistent with the results from the previous report (Supplementary Fig. 1b)[19]. These data suggest that the functional characteristics of AHG1 subfamily members are distinct from those of ABI1 subfamily members in ABA response.

We previously demonstrated that the ahg1-1ahg3-1 double mutant exhibited strong ABA-hypersensitive phenotypes in seed germination[36]. To examine the genetic and physiological relationship among AHG1 subfamily proteins in ABA response,

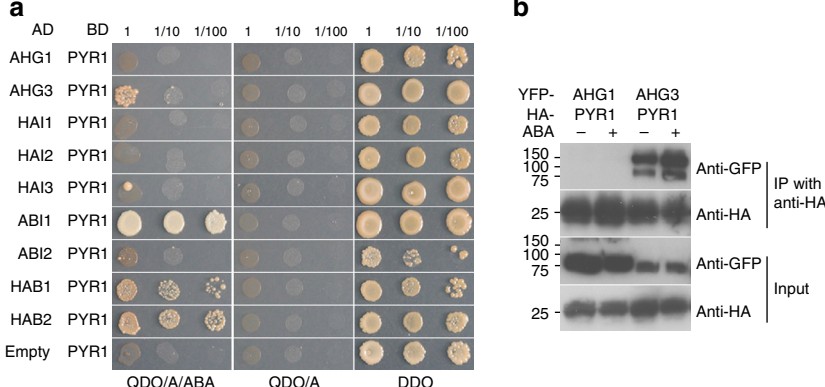

**Fig. 1** Physical interaction of group A PP2Cs with PYR1. **a** Yeast two-hybrid analysis of group A PP2Cs with PYR1. Y2H gold cells transformed with GAL4BD-PYR1 and GAL4AD-PP2Cs, as indicated. A series of tenfold serial dilutions were spotted onto DDO (Double dropout medium: SD/-Leu/-Trp), QDO/A (Quadruple dropout medium: SD/-Ade/-His/-Leu/-Trp supplemented with Aureobasidin A) and QDO/A/ABA (Quadruple dropout medium: SD/-Ade/-His/-Leu/-Trp supplemented with Aureobasidin A and ABA) medium agar plates for 7 days after inoculation. **b** Interaction of PYR1 with AHG1 and AHG3. HA-PYR1 co-immunoprecipitates with YFP-AHG3, but not YFP-AHG1, in an ABA-dependent manner. Total protein extracts from transformed *N. benthamiana* leaves were harvested 4 days after inoculation and were treated with or without 100 μM ABA for 24 h before harvesting. After co-immunoprecipitation using anti-HA matrix, the input and the immunoprecipitated samples were detected with anti-GFP and anti-HA antibodies

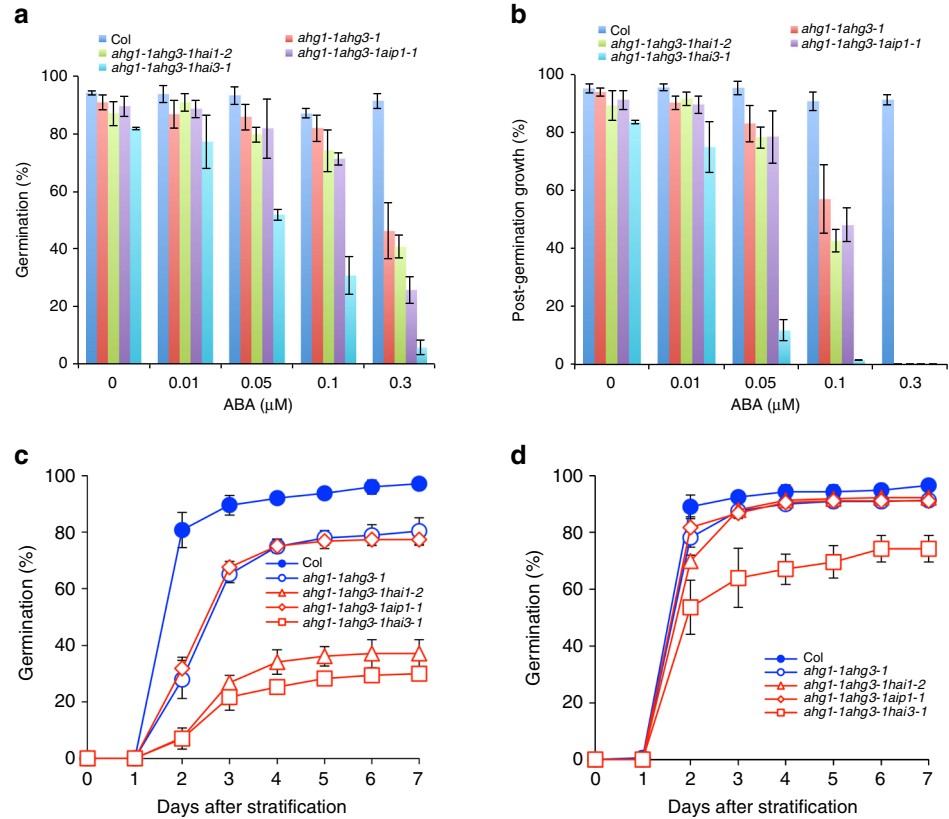

**Fig. 2** The triple mutants of *PP2C* show ABA hypersensitivity. **a, b** Germination efficiencies (**a**) and post-germination growth efficiencies (**b**) of *pp2c* double and triple mutant lines (*ahg1-1ahg3-1, ahg1-1ahg3-1hai1-2, ahg1-1ahg3-1aip1-1,* and *ahg1-1ahg3-1hai3-1*), and wild-type were examined in the presence of various concentrations of ABA at 3 days (**a**) and 7 days (**b**) after stratification. **c, d** Germination efficiencies of *pp2c* double and triple mutant lines and wild-type were examined with stratification for 0 days (**c**) or 4 days (**d**). Error bars show s.d. of three independent experiments using the same seed batch (**a–d**)

we obtained *ahg1-1ahg3-1hai1-2, ahg1-1ahg3-1aip1-1,* and *ahg1-1ahg3-1hai3-1* triple mutants. The germination (radicle emergence) and post-germination growth (seedling with expanded green cotyledons) efficiencies of *ahg1-1ahg3-1hai3-1* triple mutant were remarkably reduced in the presence of as little as 0.05 μM ABA (Fig. 2a, b). To determine whether multiple

mutations affect the response to exogenously applied ABA or seed dormancy, we tested the effect of stratification length on germination and post-germination growth efficiencies of these lines in the absence of exogenous ABA. Without stratification, the germination and post-germination growth efficiencies of the *ahg1-1ahg3-1hai1-2* and *ahg1-1ahg3-1hai3-1* triple mutants were

dramatically reduced (Fig. 2c and Supplementary Fig. 2a). Interestingly, the germination and post-germination growth efficiencies of *ahg1-1ahg3-1hai3-1* were still reduced after stratification for 4 days (Fig. 2d and Supplementary Fig. 2b). These observations indicate that these triple mutant seeds have a deeper dormancy, and further suggested that at least AHG1, AHG3, and HAI3 of the AHG1 subfamily members are involved in the regulation of seed dormancy.

**Identification of AHG1-interacting proteins.** To further address the molecular function and regulation of AHG1, we generated transgenic *Arabidopsis* plants overexpressing YFP-AHG1 (YFP-AHG1ox), HA-AHG1 (HA-AHG1ox), and YFP (YFPox) under the CaMV 35S promoter. Fluorescence microscopic analysis showed that the YFP-AHG1 proteins localized in the nucleus (Fig. 3a) in the YFP-AHG1ox lines was consistent with the data of our transient expression analysis (Supplementary Fig. 1b). We previously reported that compared to the YFPox control plants, the YFP-ABI1ox plants were small and exhibited strong ABA-insensitive phenotypes in seed germination, root growth, and stomatal responses[37]. The YFP-AHG1ox and HA-AHG1ox plants showed ABA-insensitive phenotypes in seed germination and in

root growth response as expected, whereas they showed normal plant sizes and similar ABA sensitivities in stomatal responses, compared to control plants (Fig. 3b–d and Supplementary Fig. 3a–c).

Using the YFP-AHG1ox plants, AHG1-interacting protein candidates were co-purified with YFP-AHG1. A GFP affinity column was loaded with whole-protein extracts from 3-week-old YFP-AHG1ox or YFPox plants treated with or without ABA. Western blot analysis with an anti-GFP antibody confirmed that YFP-AHG1 or YFP were purified properly (Supplementary Fig. 4a). Upon SDS-PAGE gel staining with Oriole, some visible bands specific to YFP-AHG1ox samples were detected (Supplementary Fig. 4b). Mass spectrometric analyses of three independent samples with or without ABA treatment identified proteins co-purified with the YFP-AHG1 (Supplementary Data 1). The specificity of the proteins purified by YFP affinity purification was confirmed in parallel with experiments using the YFPox plants (Supplementary Data 2). We selected the AHG1-interacting proteins that were detected in at least three YFP-AHG1ox samples, but not in all of the YFPox control samples and validated their affinity to AHG1 by yeast two-hybrid assays. As a consequence, four AHG1-interacting proteins were identified

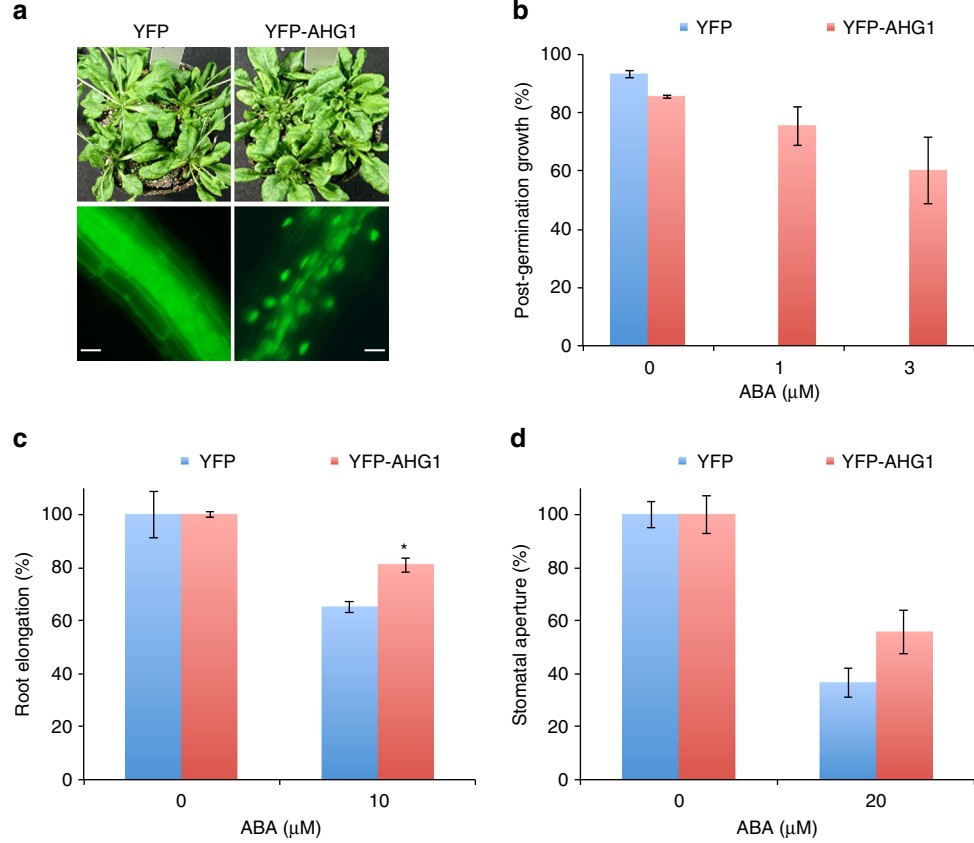

**Fig. 3** Overexpression of YFP-AHG1 causes ABA insensitivity. **a** Morphology and subcellular localization of *Arabidopsis* plants overexpressing YFP (left) and YFP-AHG1 (right) at the rosette plant stage. Plants were grown for 6 weeks in soil. Scale bars, 20 μm. **b** Post-germination growth efficiencies of overexpressing YFP and YFP-AHG1 lines in the presence of various concentrations of ABA at 7 days after stratification. Error bars show s.d. of three independent experiments using the same seed batch. **c** ABA-dependent root growth responses of overexpressing YFP and YFP-AHG1 lines. Seedlings were germinated and grown on hormone-free MS plates for 5 days and then transferred to MS plates with or without 10 μM ABA. Root length was measured 4 days after the transfer. Error bars show s.e.m. of three independent experiments. An asterisk indicates significant difference between the corresponding values (*$P < 0.05$; Tukey–Kramer test). **d** ABA-induced stomatal closure in overexpressing YFP and YFP-AHG1 lines. The epidermal tissues isolated from dark-adapted 4-to 6-week-old plants were incubated in basal buffer (5 mM MES-BTP pH 6.5, 50 mM KCl, and 0.1 mM CaCl$_2$). Pre-illuminated epidermal tissues were incubated under light (red light at 50 μmol m$^{-2}$ s$^{-1}$ and blue light at 10 μmol m$^{-2}$ s$^{-1}$) for 2.5 h with or without 20 μM ABA. Error bars show s. e.m. of three independent experiments (35 stomata per experiment and condition). No significant difference between the corresponding values ($P > 0.05$; Tukey–Kramer test)

(Supplementary Fig. 4c and Supplementary Tables 1,2). The identified AHG1-interacting proteins include a known ABA-signaling component, ABI FIVE BINDING PROTEIN 2 (AFP2), which is characterized as an ABI5-interacting protein[39,40]. The AFP family proteins have been reported to form a transcriptional co-repressor complex with TOPLESS (TPL)[41], which interestingly was identified as AHG1-interacting protein in this study. FVE/MSI4 was also identified as AHG1-interacting protein, which previously is reported to be involved in the flowering time regulation via epigenetic modifications[42]. Intriguingly, we identified DOG1 as AHG1-interacting protein, which has been shown to be a pivotal regulator of seed dormancy[12] and confirmed that DOG1 mRNA was expressed in both YFP-AHG1ox and YFPox plants (Supplementary Fig. 4d). We decided to focus on DOG1 because we suspected that AHG1 has a specific function in seed germination[36].

**Physical interaction between AHG1 and DOG1.** To clarify whether the interaction with DOG1 is specific to AHG1, we first tested the direct physical interaction between DOG1 and all nine group A PP2Cs in yeast two-hybrid assays. DOG1 could interact with all of AHG1 subfamily members, but not ABI1 subfamily members (Fig. 4a). To assess the yeast two-hybrid data, we performed co-immunoprecipitation experiments. HA-DOG1 co-immunoprecipitated with YFP-AHG1 and YFP-AHG3 in an ABA-independent manner, but not detectably with YFP-ABI1

(Fig. 4b), confirming the specific interaction between DOG1 and AHG1 subfamily members. To evaluate the biological relevance of the interaction between DOG1 and AHG1 in plant, we generated transgenic Arabidopsis plants overexpressing YFP-DOG1 (YFP-DOG1ox) that was detected in the cytoplasm and nucleus (Supplementary Fig. 5a, b). The germination and post-germination growth efficiencies of the YFP-DOG1ox lines were apparently reduced, compared to those of the YFPox control line in the presence of ABA (Fig. 4c and Supplementary Fig. 5c), indicating that DOG1 inhibits germination in an ABA-dependent manner. In contrast, the inhibitory effect of ABA on root growth and stomatal responses in the YFP-DOG1ox lines were similar to those in the control lines (Supplementary Fig. 5d, e). In contrast, the AHG1ox lines showed ABA-insensitive phenotypes in root growth (Fig. 3c and Supplementary Fig. 3b), suggesting that AHG1 and DOG1 have overlapping, but distinct physiological functions.

Based on the sequence alignment, DOG1 has five additional members in the Arabidopsis genome named DOG1-Like 1 to DOG1-Like 5 (DOGL1 to DOGL5) (Supplementary Fig. 6). We examined whether AHG1 is able to interact with all DOGLs by yeast two-hybrid assay. For DOGL5, an alternative splicing form, DOGL5.2, with higher similarity to DOG1, was used in this study. The results showed that AHG1 interacts with DOGL3 and DOGL5.2 (Supplementary Fig. 7a). DOGL3 and DOGL5.2 were also able to interact with most of the AHG1 subfamily members,

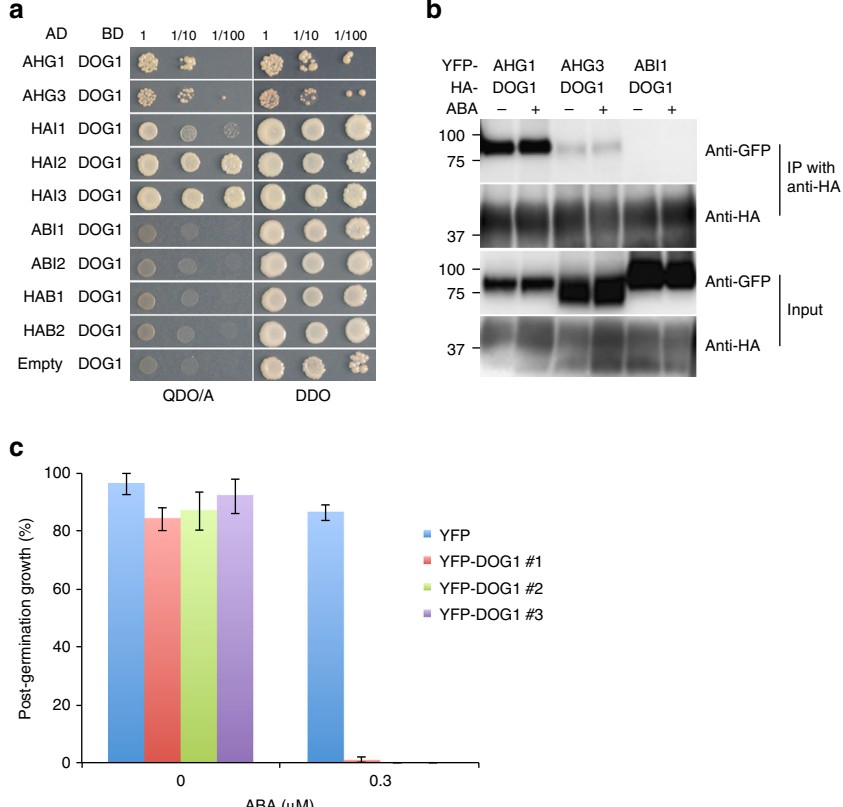

**Fig. 4** Physical interaction of AHG1 with DOG1. **a** Yeast two-hybrid analysis of group A PP2Cs with DOG1. Y2H gold cells transformed with GAL4BD-DOG1 and GAL4AD-PP2Cs, as indicated, and were spotted onto DDO (Double dropout medium: SD/-Leu/-Trp) and QDO/A (Quadruple dropout medium: SD/-Ade/-His/-Leu/-Trp supplemented with Aureobasidin A) medium agar plates for 7 days after inoculation. **b** Interaction of DOG1 with AHG1, AHG3, and ABI1. HA-DOG1 co-immunoprecipitates with YFP-AHG1 and YFP-AHG3, but not YFP-ABI1. Total protein extracts from transformed N. benthamiana leaves were harvested 4 days after inoculation and were treated with or without 100 μM ABA for 24 h before harvesting. After co-immunoprecipitation using anti-HA matrix, the input and the immunoprecipitated samples were detected with anti-GFP and anti-HA antibodies. **c** Post-germination growth efficiencies of overexpressing YFP-DOG1 and control YFP lines were treated with or without 0.3 μM ABA at 7 days after stratification. Error bars show s.d. of three independent experiments using the same seed batch

but not with ABI1 subfamily members (Supplementary Fig. 7b, c). In the transgenic *Arabidopsis* plants overexpressing YFP-DOGL3 (YFP-DOGL3ox) and YFP-DOGL5.2 (YFP-DOGL5.2ox), YFP fluorescence was detected in the cytoplasm and the nucleus (Supplementary Fig. 8a, b). The YFP-DOGL3ox lines exhibited lower germination and post-germination growth efficiencies similar to the YFP-DOG1ox line, when compared to the YFPox control line in the presence of ABA (Supplementary Fig. 8c, d). Surprisingly, the YFP-DOGL5.2ox lines did not show an ABA-hypersensitive phenotype in seed germination (Supplementary Fig. 8c, d), suggesting that DOGL5.2 may not function like DOG1, even though it has the ability to interact with AHG1.

**Genetic and functional interactions between AHG1 and DOG1.** To investigate the genetic relationship between *AHG1* and *DOG1*, we constructed *ahg1-1dog1-2* and *ahg1-1dog1-3* double mutants, and transgenic plants overexpressing both HA-AHG1 and YFP-DOG1 fusion proteins (Supplementary Fig. 9a). The germination and post-germination growth efficiencies of *ahg1-1dog1-2* and *ahg1-1dog1-3* double mutants were apparently reduced in the presence of ABA, similar to those of *ahg1-1* (Fig. 5a and Supplementary Fig. 9b). Correspondingly, the HA-AHG1ox/YFP-DOG1ox double-expression line showed a strong ABA-insensitive phenotype, similar to the HA-AHG1ox line (Fig. 5b and Supplementary Fig. 9c). These results suggest that

DOG1 functions upstream of AHG1 in the ABA signaling pathway, and led us to the idea that DOG1 directly regulates the PP2C activity of AHG1 in an ABA-dependent manner. To test this hypothesis, we examined the phosphatase activity of recombinant truncated AHG1 in the presence or absence of recombinant DOG1 or ABA using a synthetic phosphopeptide[43], corresponding to the regulatory phosphorylation site of SnRK2s (HSQPK(pS)TVGTP) and an artificial substrate[44] for phosphatase 2A, 2B, and 2C (RRA(pT)VA) in vitro. Interestingly, DOG1 impaired the PP2C activity of truncated AHG1 for synthetic SnRK2s phosphopeptide, but not for the artificial substrate, regardless of ABA in our in vitro assay conditions (Fig. 5c, Supplementary Fig. 10). These findings imply that DOG1 regulates the activation state of SnRK2s through the inhibition of the PP2C activity of AHG1.

**The N-terminal portion of DOG1 interacts with AHG1.** To determine the regions of DOG1 required for AHG1 interaction, various deleted forms of DOG1 fused to YFP were constructed and their ability to interact with AHG1 was examined (Supplementary Fig. 11a, b). Since some deleted forms of DOG1 that excluded a conserved region were difficult to express (Supplementary Fig. 11b; Lane 3,4,5), we could not evaluate the results in those samples. HA-AHG1 co-immunoprecipitated with YFP-DOG1$_{\Delta 257-291}$, but not detectably or very weakly with YFP-

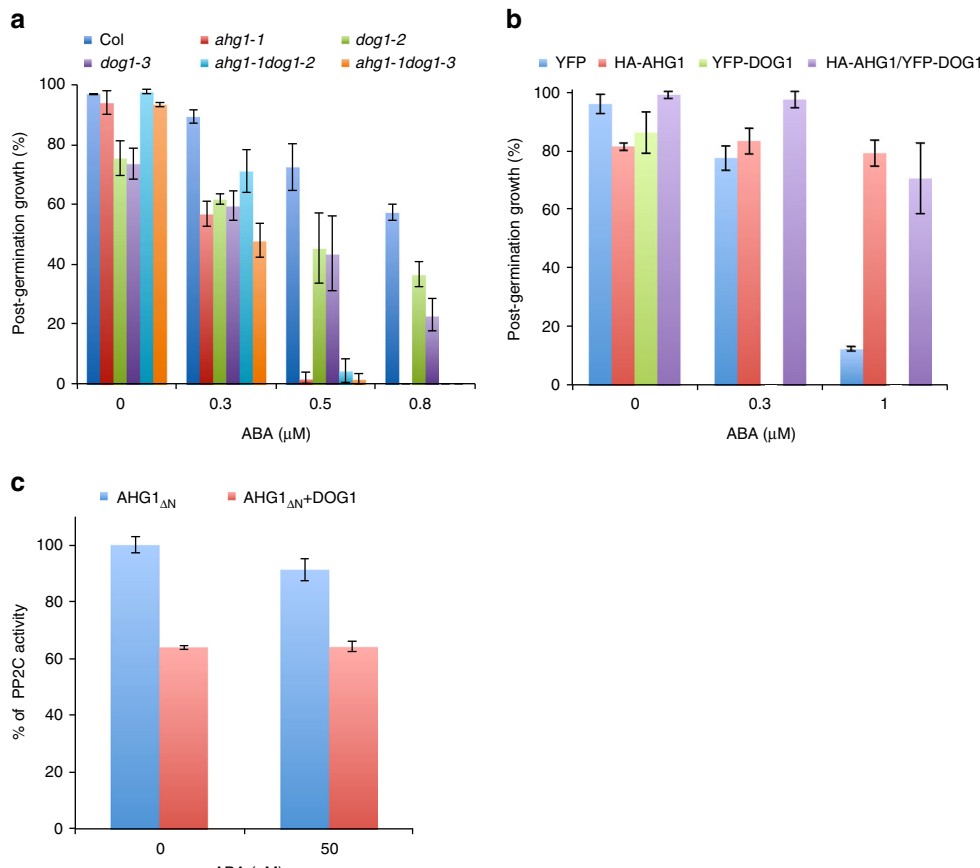

**Fig. 5** Genetic and functional interactions between AHG1 and DOG1. **a** Post-germination growth efficiencies of the single and double mutant lines (*ahg1-1*, *dog1-2*, *dog1-3*, *ahg1-1dog1-2*, and *ahg1-1dog1-3*) and wild-type were examined in the presence of various concentrations of ABA at 7 days after stratification. **b** Post-germination growth efficiencies of the double overexpressing (HA-AHG1ox/YFP-DOG1ox), parental (HA-AHG1ox and YFP-DOG1ox), and control YFP lines were examined in the presence of various concentrations of ABA at 7 days after stratification. Error bars show s.d. of three independent experiments using the same seed batch (**a**, **b**). **c** The PP2C activities of the truncated AHG1$_{\Delta N}$ were measured with or without DOG1 or ABA using the synthetic phosphopeptide, corresponding to the regulatory phosphorylation site of SnRK2s (HSQPK(pS)TVGTP) as a substrate. Error bars show s.d. of three independent experiments

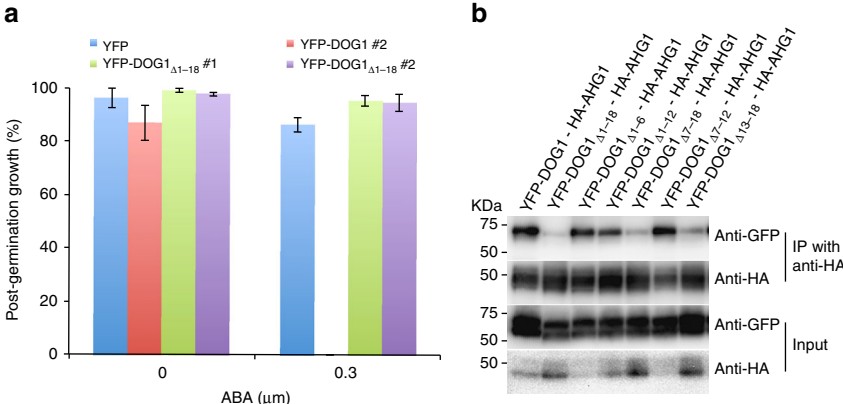

**Fig. 6** Physical interaction of DOG1 with AHG1 affects DOG1 function. **a** Post-germination growth efficiencies of overexpressing YFP-DOG1$_{\Delta1-18}$, YFP-DOG1, and control YFP lines were treated with or without 0.3 μM ABA at 7 days after stratification. Error bars show s.d. of three independent experiments using the same seed batch. **b** Interaction of the N-terminal portion of DOG1 with AHG1. HA-AHG1 co-immunoprecipitates with the deleted forms of YFP-DOG1. Total protein extracts from transformed *N. benthamiana* leaves were harvested 5 days after inoculation. After co-immunoprecipitation using anti-HA matrix, the input and the immunoprecipitated samples were detected with anti-GFP and anti-HA antibodies

DOG1$_{\Delta1-18}$ (Supplementary Fig. 11b). In the transgenic *Arabidopsis* plants overexpressing YFP-DOG1$_{\Delta1-18}$ (YFP-DOG1$_{\Delta1-18}$ox), YFP fluorescence was detected in the cytoplasm and the nucleus (Supplementary Fig. 12a, b). The germination and post-germination growth efficiencies of these lines were not apparently reduced, compared to those of the YFP-DOG1 line in the presence of ABA, and were similar to those of the YFPox control line (Fig. 6a and Supplementary Fig. 12c), suggesting that the N-terminal 18 residues are necessary for the DOG1 function, and further implying that the interaction with AHG1 is important for DOG1 function. To further narrow the region required for interaction with AHG1, additional deleted forms of YFP-DOG1 were constructed and their abilities to interact with AHG1 were examined. HA-AHG1 co-immunoprecipitated with YFP-DOG1$_{\Delta1-6}$, YFP-DOG1$_{\Delta1-12}$, and YFP-DOG1$_{\Delta7-12}$, but not detectably or very weakly with YFP-DOG1$_{\Delta7-18}$ and YFP-DOG1$_{\Delta13-18}$, like YFP-DOG1$_{\Delta1-18}$ (Fig. 6b). Thus, we concluded that the six-residue sequence of DOG1 spanning position 13–18, DSYLEW, is essential for interacting with AHG1 (Fig. 6b).

**DOG1 is an α-helical heme-binding protein**. To further address the function of DOG1, we expressed the recombinant DOG1 in *Escherichia coli* under two different conditions, short-time (16 h) expression in LB medium (condition I) and long-time (50–60 h) expression in an enriched TB medium (condition II). Interestingly, reddish-brown colored DOG1 was obtained from cells under condition II, while colorless DOG1 was obtained from cells under condition I, suggesting that the colored DOG1 may have the potential to bind to a small chromophore molecule such as heme (Fig. 7a). The absorption spectrum of the colored DOG1 exhibited characteristics of heme protein complex peaks, δ peak at 360 nm, γ (Soret) peak at 425 nm, β peak at 543 nm, and α peak around 575 nm appeared as a shoulder of β peak (Fig. 7b). These peaks are consistent with a typical hexacoordinate low spin Fe (III) heme. Analyzing the spectrum as described in the Methods revealed that about 70% of the colored DOG1 expressed in *E. coli* under condition II was bound to heme. The heme binding ability of DOG1 was further demonstrated by titration of hemin to the colorless apo-DOG1 expressed in *E. coli* under condition I (Fig. 7c and Supplementary Fig. 13a, b). For nonlinear curve fitting of the data, an increase in the Soret peak against hemin concentration was fit to a 1:1 stoichiometric heme binding, and the $K_d$ values were 59 nM for the untagged DOG1 (Fig. 7d) and

84 nM for the N-terminal His$_6$-tagged DOG1 (Supplementary Fig. 13f and Supplementary Table 3).

To understand the secondary structure of DOG1, we performed Far-UV circular dichroism (CD) analysis. The apo-DOG1 showed a spectrum with negative peaks at 222 and 208 nm and a positive peak at 193 nm (Supplementary Fig. 14a), indicating that DOG1 is a typical α-helical protein. This result is consistent with the secondary structures predicted by Jpred4 and Phyre2[45,46] (Supplementary Fig. 6). The CD spectrum of the heme-bound DOG1 is nearly superimposable with that of the apo-DOG1, suggesting that heme coordination does not affect the secondary structure of DOG1, but may induce tertiary structural changes.

**Heme-binding site is essential for DOG1 function**. According to Li et al.[11], five different amino acids, His, Met, Cys, Tyr, and Lys can preferentially function as axial ligands to heme, and histidine is the dominant residue (ca. 80%). To confirm the possible involvement of histidine residues in the binding of heme, we made mutant DOG1 proteins (H39A, H71A, H153A, H245A, and H249A) in which each of the five histidine residues were substituted with alanine and measured their electronic absorption spectra. The wild-type and mutant DOG1 proteins in apo forms were incubated with an excess of hemin and passed through a gel filtration column to remove the unbound hemin. This treatment produced the fully heme-bound, reddish-brown colored wild-type and mutant DOG1 proteins without free hemin. All the DOG1 mutants, except for H245A mutant, exhibited the similar spectral characteristics as the wild-type DOG1 (Fig. 8a, b). Interestingly, DOG1$^{H245A}$ showed drastic spectral changes with the appearance of new peaks around 390 and 645 nm (Fig. 8a). These peaks are a characteristic of the pentacoordinate high-spin heme-iron Fe$^{3+}$ form, indicating that His245 could function as an axial ligand for bound heme in wild-type DOG1 (Fig. 8a). However, the characteristic peaks for the hexacoordinate low-spin heme at 360 and 420 nm were still observed for DOG1$^{H245A}$, albeit at lower intensities than the wild-type DOG1.

According to the secondary structure predicted by the Jpred4 and Phyre2 programs, His249 and His245 are located close to each other in the same α-helix (Supplementary Fig. 6). To test whether His249 is an axial ligand for the hexacoordinated form in the DOG1$^{H245A}$, we measured the spectral properties of the double amino acid substitution (H245A and H249A) DOG1 mutant protein. The fully heme-bound, reddish-brown colored

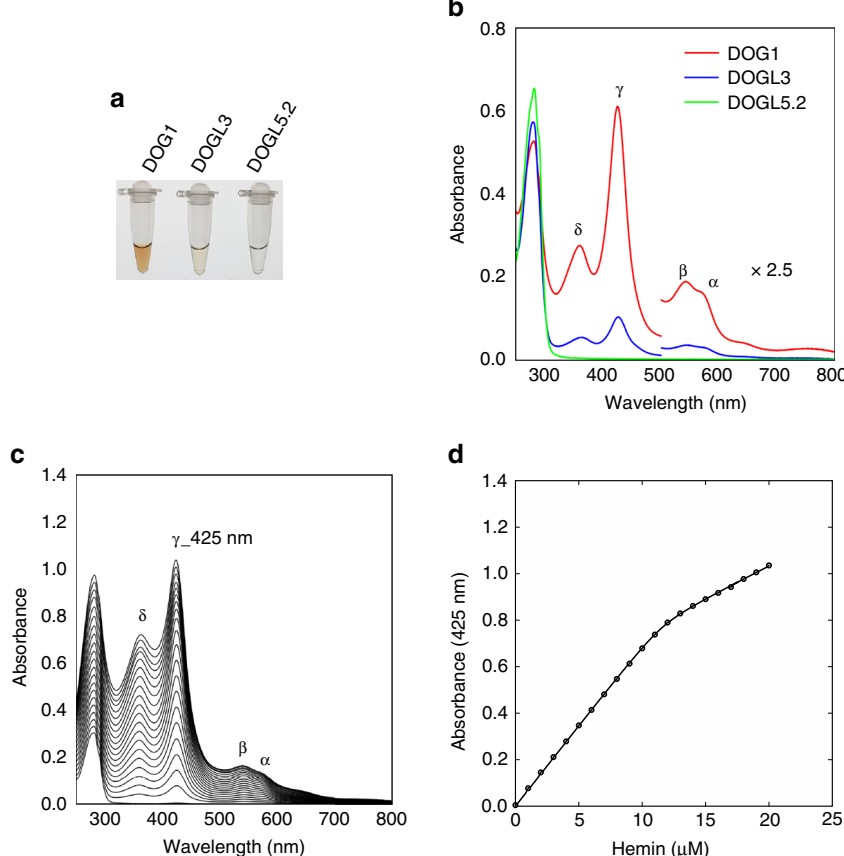

**Fig. 7** Heme binding properties of DOG1. **a** *E. coli* expressed N-terminal His$_6$-tagged DOG1 (29 μM), DOGL3 (40 μM), and DOGL5.2 (40 μM) for 50-60 h in TB medium (condition II). **b** Electronic absorption spectra of the DOG1 (11 μM), DOGL3 (15 μM), and DOGL5.2 (15 μM) shown in **a**. **c** Electronic absorption spectra of the untagged DOG1 (9.4 μM) after the addition of hemin at the amount of up to ca. 2 mol equivalents of protein. **d** The γ (Soret) peak absorbance at 425 nm plotted as a function of hemin concentration. Nonlinear curve fitting of experimental data as described in the Methods produced $K_d$ = 59 nM, $\varepsilon_{DH}$ = 79.4 mM$^{-1}$ cm$^{-1}$, $\varepsilon_H$ = 27.4 mM$^{-1}$ cm$^{-1}$, and $x$ = 0.81. $\varepsilon_{DH}$, and $\varepsilon_H$ are the extinction coefficients of the DOG1-bound heme and the free hemin at 425 nm, respectively, while "$x$" represents a fraction of the active hemin, which can be incorporated into DOG1

DOG1$^{H245AH249A}$ showed a dramatically perturbed electronic absorption spectrum, indicating that a pentacoordinate high-spin heme is dominant in DOG1$^{H245AH249A}$ (Fig. 8c). This result strongly suggests that the new axial ligand is His249 in the hexacoodinate low-spin heme of DOG1$^{H245A}$ (Supplementary Fig. 14b). These data suggest that two histidine residues, His245 and His249, would be located in close proximity to each other and would act as an alternative axial ligand in wild-type DOG1. In support of this idea, DOG1$^{H245A}$ expressed under condition II showed a faint red color, while DOG1$^{H245AH249A}$ was almost colorless (Supplementary Fig. 14c). As shown in Supplementary Fig. 14d, the absorption spectra of these samples depicted in Supplementary Fig. 14c are largely different from those for the fully heme-bound, reddish-brown colored forms of the corresponding mutants shown in Fig. 8c. Hemin titration experiments provided $K_d$ values of 129 nM for DOG1$^{H245A}$ and 918 nM for DOG1$^{H245AH249A}$ (Supplementary Fig. 13 and Supplementary Table 3), further confirming their lower heme-binding affinities than wild-type DOG1. It is noteworthy that DOG1 lacking the N-terminal 18 residues, which is required for the interaction with AHG1 (Fig. 6a, b), showed a reddish-brown color and spectral characteristics similar to wild-type DOG1 (Supplementary Fig. 14c, d). When HA-AHG1 was co-expressed with either YFP-DOG1, YFP-DOG1$^{H245A}$ or YFP-DOG1$^{H245AH249A}$ protein in *N. benthamiana*, HA-AHG1 co-immunoprecipitated with YFP-DOG1, YFP-DOG1$^{H245A}$, and YFP-DOG1$^{H245AH249A}$ at very similar efficiencies, regardless of hemin (Supplementary

Fig. 14e). These data suggest that the interaction with AHG1 of DOG1 is independent of its heme coordination.

YFP-DOG1ox and YFP-DOGL3ox lines showed a strong ABA-hypersensitive phenotype in seed germination, while YFP-DOGL5.2ox lines did not (Fig. 4c and Supplementary Fig. 8d), even though all three proteins had the ability to interact with AHG1. According to the sequence alignment, histidine residues His245 and His249, which bind to heme in DOG1, are conserved in DOGL3, but not in DOGL5.2 (Supplementary Fig. 6). Recombinant DOGL3 expressed under condition II was faint red in color (Fig. 7a). Analysis of the electronic absorption spectrum revealed that heme content in the colored DOGL3 was ca. 8% (Fig. 7b). In contrast, recombinant DOGL5.2 expressed under the same condition was colorless (Fig. 7a, b) and heme binding was not observed, even when it was treated with excess hemin. These results strongly suggest that the heme-binding abilities of DOG1, DOGL3, and DOGL5.2 are correlated well to their abilities to ABA-hypersensitive phenotype in seed germination (Fig. 4c and Supplementary Fig. 8d).

To see whether the heme binding via His245 and His249 is essential for DOG1 function in plants, we generated transgenic *Arabidopsis* plants overexpressing YFP-DOG1$^{H245A}$ (YFP-DOG1$^{H245A}$ox) and YFP-DOG1$^{H245AH249A}$ (YFP-DOG1$^{H245A-H249A}$ox) that were detected in the cytoplasm and the nucleus (Supplementary Figs 15a, b and 16a, b). The YFP-DOG1$^{H245A}$ox lines exhibited lower germination and post-germination growth efficiencies similar to YFP-DOG1ox line, compared to the YFPox

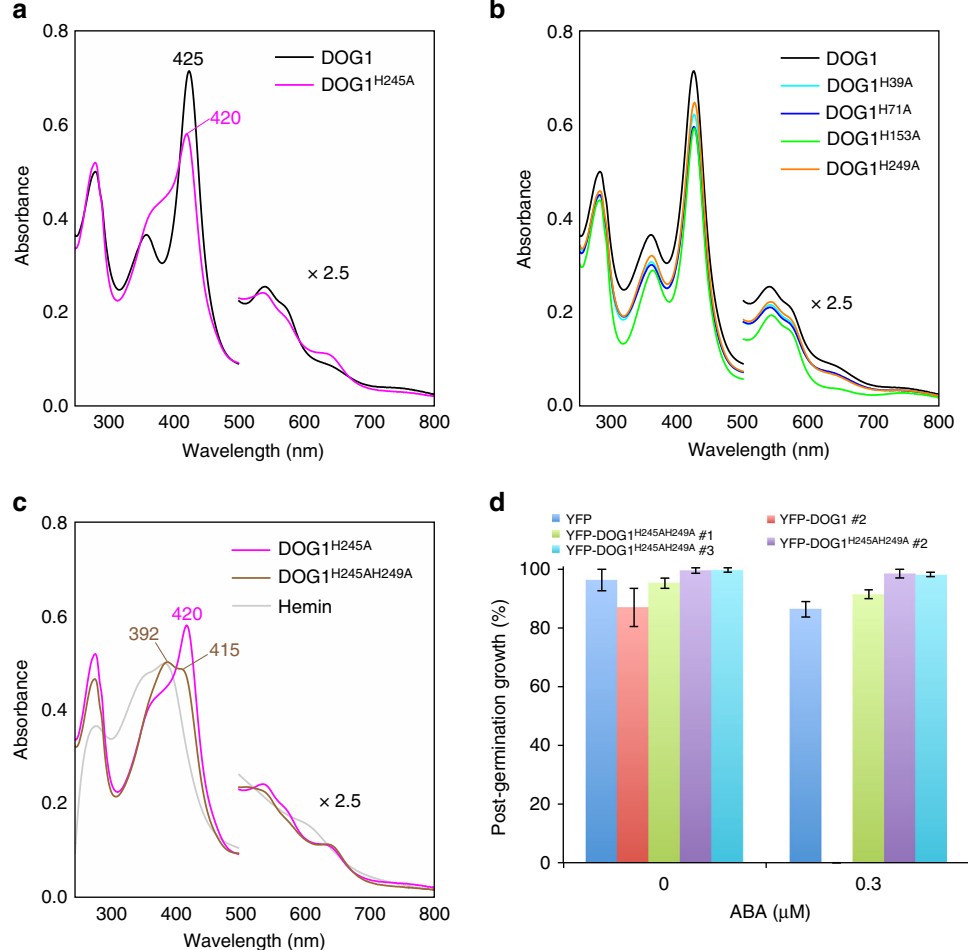

**Fig. 8** An alternative axial ligand to heme in the wild-type DOG1. **a** Comparison of electronic absorption spectra of the fully heme-bound wild-type DOG1 and its H245A single mutant. **b** Overlay of electronic absorption spectra of the fully heme-bound wild-type DOG1 and its His-to-Ala single mutants (H39A, H71A, H153A, and H249A). **c** Comparison of electronic absorption spectra of the fully heme-bound H245A single mutant DOG1 and H245AH249A double mutant DOG1, as well as free hemin. The fully heme-bound wild-type and mutant DOG1 proteins were prepared by incubating their apo forms with an excess of hemin, followed by a gel filtration column to remove the unbound hemin. All the spectra shown in **a–c** were measured for the N-terminal His6-tagged proteins at concentration of ca. 8 μM. **d** Post-germination growth efficiencies of overexpressing YFP-DOG1^H245AH249A, YFP-DOG1, and control YFP lines were treated with or without 0.3 μM ABA at 7 days after stratification. Error bars show s.d. of three independent experiments using the same seed batch

control line in the presence of ABA (Supplementary Fig. 15c,d). In contrast, the YFP-DOG1^H245AH249Aox lines showed similar germination and post-germination growth efficiencies to the YFPox control line in the presence of ABA (Fig. 8d and Supplementary Fig. 16c). These in vivo results correlate well with the in vitro spectroscopic results and support the biological relevance of heme binding for DOG1 function in ABA signaling.

## Discussion

The solved crystallographic structures of ABA-PYL/RCAR-PP2Cs complexes revealed that a conserved tryptophan residue in PP2Cs, excluding AHG1, inserted in the entrance of the internal cavity of PYL/RCARs formed a water-mediated hydrogen-bond network with the ABA-bound PYL/RCARs complex[47–49]. AHG1 lacks this conserved tryptophan residue and is therefore predicted to be unable to interact with ABA-bound PYL/RCARs[20,50]. This idea is supported by the fact that none of the PYL/RCAR ABA receptors were identified as AHG1-interacting proteins in this study (Supplementary Data 1 and Supplementary Tables 1, 2). These observations led us to believe that AHG1 functions in a unique regulatory system that is independent of PYL/RCARs.

To examine the AHG1-specific regulatory system, we successfully identified four AHG1-interacting proteins (DOG1, AFP2, TPL, and FVE/MSI4) (Supplementary Fig. 4c), and characterized the unique properties in the interactions with AHG1, AHG3, and ABI1. Among these AHG1-interacting proteins, DOG1 was particularly enticing because AHG1 functions primarily in the seeds[36]. DOG1 is one of the pivotal regulators of seed dormancy and germination, although its molecular function is largely unknown[2–4]. A recent study has shown that DOG1 also regulates flowering time through microRNA, suggesting that DOG1 is expressed and functions in mature plants (Supplementary Fig. 4d)[51]. Interestingly, AHG1 subfamily members interact with DOG1, while no other ABI1 subfamily members do (Fig. 4a, b). We also demonstrate the clear genetic and functional interactions between *AHG1* and *DOG1* loci (Fig. 5). In addition, removing the N-terminal portion of DOG1 required for the AHG1-interaction impaired the ability of DOG1 to confer an ABA-hypersensitive phenotype in seed germination, indicating that the interaction with AHG1 is indispensable for DOG1 function (Fig. 6). Therefore, it is likely that AHG1 and DOG1 constitute an alternative regulatory system, distinct from the PYL/RCAR-PP2C regulatory system, to control seed dormancy and

germination. Since *AHG1* and *DOG1* homologs are conserved among higher plants[32,52,53], the DOG1-AHG1 regulatory system may be common. Recently, the same region of DOG1 responsible for AHG1 interaction had been reported to be a self-dimerization site[54]. While it remains unclear how self-dimerization is involved in DOG1 function, DOG1 might be a core protein in a large protein complex, regulating components including itself and AHG1. It is noteworthy that the HA-AHG1 levels seemed to be less accumulated when co-expressed with the deleted forms of YFP-DOG1 that can interact with HA-AHG1, in comparison with deleted forms of YFP-DOG1 that cannot interact with HA-AHG1 (Fig. 6b). Although we could not exclude the possibility that this is caused by the experimental conditions, DOG1-interaction might affect the AHG1 level.

Epistatic analysis suggested that DOG1 functions upstream of AHG1 in ABA signaling in seed germination (Fig. 5). AHG1 was reported to interact with SnRK2.3 in vivo and regulate the activation state of OST1/SnRK2.6[19,20]. In our investigation, DOG1 could reduce the PP2C activity of AHG1, regardless of ABA in vitro with a synthetic SnRK2s phosphorylation site peptide, but not with a conventional PP2C substrate (Fig. 5c, Supplementary Fig. 10). We thought that if DOG1 negatively regulates the AHG1 PP2C activity, the HA-AHG1ox/YFP-DOG1ox line would show a similar phenotype to the YFP-DOG1ox line. Indeed, the PYL5/HAB1 double expression lines exhibited a similar phenotype to the PYL5ox expression line, indicating that the PYL5 ABA receptor prohibited HAB1 function[38]. However, the HA-AHG1ox/YFP-DOG1ox double expression line showed an ABA-insensitive phenotype similar to the HA-AHG1ox line (Fig. 5b). Presumably, the over-production of HA-AHG1 overcame the inhibitory effect of the over-produced YFP-DOG1. It is possible that DOG1 needs additional modifications or interactions with other components for its function. Such a modification or constructing a large complex with multicomponents might be required for the proper DOG1 regulation of AHG1.

The DOG1-AHG1 complex may also regulate other interactors of AHG1. One good candidate is ABI5, which is one of the major determinants of seed germination regulation[55,56]. We identified AFP2 and TPL as AHG1-interacting proteins (Supplementary Fig. 3c). AFP negatively regulates ABI5 controlling its protein levels through the ubiquitin-mediated system[39], and is also reported to interact with TPL to form a transcriptional co-repressor complex[41]. Recent studies indicated that DOG1 affects the expression level of *ABI5*, *AFPs*, and some group A *PP2C* genes[14]. ABI5 is one of the well-known major substrates of subgroup III SnRK2s[22,25], and both of these components are also reported to interact with AHG1[19,57]. Taken together, these observations suggest that the DOG1-AHG1 complex is likely to regulate ABI5 function through SnRK2s and AFPs-TPL repressor complex.

Although AHG3 and other AHG1 subfamily members have been reported to interact with some of PYL/RCAR ABA receptors[7], they all are able to interact with DOG1 (Fig. 4a). Thus these members function like AHG1 in the DOG1-dependent system. This idea is supported by the data that the triple loss-of-function mutants of these PP2Cs showed strong ABA hypersensitivity and increased the seed dormancy phenotypes in seeds (Fig. 2). Their ability to interact with both DOG1 and PYL/RCAR ABA receptors indicates that these PP2Cs can be important hub regulators connecting the two distinct regulatory pathways to integrate multiple signals and fine-tune the regulation of seed dormancy and germination.

In this study, we demonstrated that DOG1 binds to heme (Fig. 7), and inferred that histidine residues function as an axial ligand for the bound heme. Conversion of histidine residues to alanines abolished the DOG1 activity to confer the ABA-

hypersensitive phenotype in germination (Fig. 8), strongly supporting the importance of heme coordination for DOG1. Heme is a well-known key molecule that regulates diverse biological activities[10,11]. Some heme-binding proteins are reported to function as sensors for oxygen and nitric oxide (NO)[11], including the widely studied mammalian NO sensor, Guanylate cyclase[58,59], and FixL, a nitrogen-fixing bacteria which is a well-known oxygen sensor[60]. Reactive oxygen species (ROS) and NO counteract ABA to change the phase transition from seed dormancy to germination[61,62]. Similarly, heme-binding DOG1 might monitor the ROS and the NO levels affected by physiological or developmental stimuli in seeds. Indeed, DOG1 has been shown to undergo post-translational modifications (PTMs) during after-ripening[13], implying that DOG1 is regulated by PTMs related to ROS and NO, such as cysteine oxidation and S-nitrosylation[2,62], and might be the hub regulator integrating environmental signals. In addition, there are reports showing a link between heme and ABA response. The heme scavenger tryptohan-rich sensory protein (TSPO) is principally detected in dry seeds, and is accumulated under ABA treatments in *Arabidopsis*, and TSPO overexpression lines show a weak ABA-hypersensitive phenotype in seed germination[8,63].

The results presented here, along with the previous studies, lead us to propose a model as a working hypothesis for the regulation of seed dormancy and germination (Fig. 9). As previously reported, ABA directly regulates PYL/RCAR-PP2C and downstream components such as SnRK2s and ABI5, which in turn control seed dormancy and germination. The present research points to the hypothesis that DOG1 and AHG1 constitute another parallel regulatory pathway, in which DOG1 integrates environmental signals or physiological conditions other than ABA, and the DOG1-AHG1 complex regulates downstream components including SnRK2s and ABI5. Presumably, the downstream components of these two regulatory pathways overlap. In addition, there are several PP2Cs that appear to function in both the pathways, suggesting that both the intense cross-talk and the signal integration between the two pathways are essential for the regulation of seed dormancy and germination to maximize the environmental adaptability of the plant life cycle.

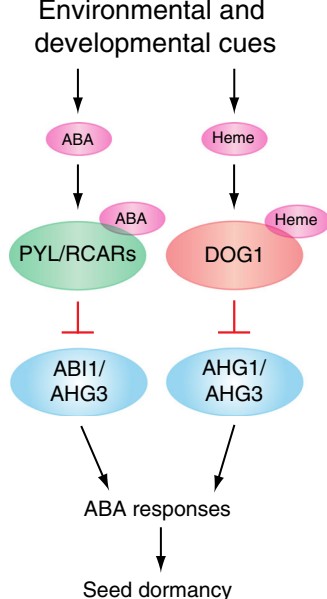

**Fig. 9** A hypothesized regulatory system of seed dormancy. See the text for detail

We note that, while a prior version of this manuscript was under revision, the physical and genetic interaction of DOG1 with PP2Cs was reported[64]. The conclusion of that study, that PP2Cs interact with and are epistatic to DOG1, is consistent with the current work. The reduction of PP2C activity of AHG1 and heme-binding data here provide further biochemical details regarding the function of DOG1.

## Methods

**Plant materials and growth conditions.** *Arabidopsis thaliana* wild-type Columbia (Col), *ahg1-1* (see ref. [35]), *ahg3-1* (see ref. [33]), *hai1-2* (SALK_1082282), *aip1-1* (SALK_090738), *hai3-1* (SALK_033011), *dog1-2* (see ref. [12]), and *dog1-3* (SALK_000867) were used in this study. Plants were grown on MS plates containing 1× Murashige and Skoog salt mix, 1% sucrose, 2.5 mM MES (pH5.8), and 0.8% agar or on soil at 22 °C under 16 h light/8 h dark cycles. Seeds were first stratified at 4 °C for 4 days and then transferred to a growth chamber, unless otherwise indicated.

**Germination and growth assays.** For germination assays, approximately 50 seeds were sown on MS plates containing various concentration of ABA (Sigma). Germination (radicle emergence) and post-germination growth (seedling with expanded green cotyledons) were scored daily for 3 days and 7 days. For seed dormancy assays, approximately 50 seeds were sown on hormone-free MS plates. Germination (radicle emergence) and post-germination growth (seedling with expanded green cotyledons) were scored daily for 7 days. For root growth inhibition assays, 12–15 seedlings were germinated and grown on hormone-free MS plates for 5 days and then transferred to MS plates containing various concentration of ABA (Sigma) for 4 days. For ABA-induced stomatal closure assays, stomatal apertures in the abaxial epidermis were measured microscopically[65]. The epidermal tissues isolated from dark-adapted 4- to 6-week-old plants were incubated in basal buffer (5 mM MES-BTP pH 6.5, 50 mM KCl, and 0.1 mM CaCl$_2$). The epidermal tissues were pre-illuminated under light [blue light (Stick-B-32; EYELA) at 10 μmol m$^{-2}$ s$^{-1}$ superimposed on background red light (LED-R; EYELA) at 50 μmol m$^{-2}$ s$^{-1}$] for 2.5 h, then incubated with or without 20 μM ABA for 2.5 h under light (same as above).

**Vector constructions.** Full-length *ABI1*, *ABI2*, *HAB1*, and *PYR1* cDNAs were previously cloned into entry vector[16] (Invitrogen). Full-length *AHG1*, *AHG3*, *HAI1*, *HAI2/AIP1*, *HAI3*, *DOG1*, *DOGL1*, *DOGL2*, *DOGL3*, *DOGL4*, and *AFP2* cDNAs were cloned into pENTR vector (Invitrogen) and sequenced. The entry clones, which are inserted into full-length *HAB2* (CIW00104), *TPL* (G09916), *DOGL5.2* (PENTR221-AT3G14880), *FVE/MSI4* (U15529) cDNAs, were ordered from ABRC and sequenced. Site-directed mutagenesis was performed with the KOD-Plus-mutagenesis kit (TOYOBO) using entry clones as templates. The entry clones were transferred to the destination vectors pACTGW-attR, pASGW-attR, pH35YG, and pEarleyGate 201 by Gateway LR clonase II enzyme mix recombination reaction.

**YFP fusion protein expression analyses.** Full-length group A PP2Cs cDNAs were cloned into pH35YG. *Agrobacterium tumefaciens* strain GV3101 carrying the gene of interest was used and infiltrated at an OD$_{600}$ of 0.5 together with p19 strain in *N. benthamiana*. Mesophyll protoplasts were isolated from leaves after 5 days of infiltration, according to instructions[66]. Fluorescence imaging was analyzed by confocal microscopy (Nikon Eclipse TE2000-U) using 488 nm excitation and 500–550 nm emission filters for YFP.

**GFP expression and antiserum preparation.** *sGFP* gene was cloned into the *NdeI/XhoI* sites of the expression vector pET28a (+) (Novagen) and transformed into *E. coli.* strain BL21 Rossetta2 (DE3) (Merck). The *E.coli.* cells carrying expression plasmid were grown at 37 °C, and protein expression was induced by the addition of IPTG to 1 mM at OD$_{600}$ 0.6–0.7 in LB medium. After a 3 h incubation, the cells were harvested by 4000×*g* centrifugation for 10 min at 4 °C and the pellet was resuspended in the sonic buffer (50 mM sodium phosphate buffer pH 7.0, 300 mM NaCl, 10 mM imidazole, EDTA-free protease inhibitor cocktail (Roche)). Cells were sonicated on ice seven times for 30 s with regular resting intervals. A supernatant was obtained after centrifugation at 20,000×*g* for 10 min at 4 °C and incubated with TALON metal resin (Clontech) for 30 min at 4 °C. After being washed three times in the sonic buffer without the protease inhibitor cocktail, the matrix was resuspended with an elution buffer (50 mM sodium phosphate buffer pH 7.0, 300 mM NaCl, 250 mM imidazole) and incubated for 30 min. The protein was further purified with Sephacryl S-100 (GE Healthcare) using a solution of PBS buffer. Antisera against GFP were generated by Scrum Inc. and were evaluated by immunoblot for further use in this study as anti-GFP antibody.

**The construction of transgenic *Arabidopsis* lines.** Full-length *AHG1*, *DOG1* (WT and mutants), *DOGL3*, and *DOGL5.2* cDNAs were cloned into pH35YG and full-length *AHG1* cDNA was cloned into pEarleyGate 201. *Agrobacterium tumefaciens*

strain GV3101 carrying the gene of interest was used to transform wild-type Col-0 with plants by the floral dip method[67]. Transgenic plants were screened for hygromycin or bialaphos resistance, and the homozygous T3 lines were isolated. Subcellular localizations of YFP-AHG1, YFP-DOG1, YFP-DOGL3, and YFP-DOGL5.2 in the root were analyzed by fluorescence microscope (Nikon Eclipse N*i*). Transgenic plants were verified by immunoblotting using a 1:10,000 diluted anti-GFP antibody (in this study), followed by a 1:50,000 diluted anti-Rabbit HRP-conjugate (Pierce #31460). Detection was performed using SuperSignal West Dura extended Duration Substrate (Pierce), according to the manufacturer's instructions. The uncropped full-scan data of immunoblots and gel are shown in Supplementary Fig. 17.

**Yeast two-hybrid assay.** The Matchmaker Gold Yeast two-hybrid system (Clontech) was used according to the manufacturer's instruction. Full-length *PYR1*, group A *PP2Cs*, *DOG1/DOGLs*, and *AHG1-interacting proteins* cDNAs were cloned into pACTGW-attR and pASGW-attR. Each vector was transformed into yeast Y2H gold strain (Clontech). The transformed yeast cells were sprayed onto DDO (Double dropout medium: SD/-Leu/-Trp) medium agar plates and incubated at 30 °C for 5 days. A series of tenfold serial dilutions were spotted onto DDO, QDO/A (Quadruple dropout medium: SD/-Ade/-His/-Leu/-Trp supplemented with Aureobasidin A), and QDO/A/ABA (Quadruple dropout medium: SD/-Ade/-His/-Leu/-Trp supplemented with Aureobasidin A and 10 μM ABA) medium agar plates for 7 days after inoculation.

**Co-immunoprecipitation experiments in *N. benthamiana*.** Full-length *AHG1*, *AHG3*, *ABI1*, and *DOG1* (WT and mutants) cDNAs were cloned into pH35YG, and the full-length *PYR1*, *AHG1*, and *DOG1* cDNAs were cloned into pEarleyGate 201. *A. tumefaciens* strain GV3101 carrying the gene of interest was used and infiltrated at an OD$_{600}$ of 0.5, together with p19 strain in *N. benthamiana*. After 4 days of infiltration, the infiltrated leaves were sprayed with water containing 0.01% Silwet L-77 (BMS) and either 100 μM ABA or 50 μM hemin for 24 h prior to leaf excision. For protein extraction, *N. benthamiana* leaves (0.75 g) were harvested and ground to a powder in liquid nitrogen. Ground tissues were resuspended into 1.5 mL of extraction buffer (50 mM Na-phosphate pH 7.4, 150 mM NaCl, 0.1% NP-40, 1 mM DTT and 1× protease inhibitor cocktail (Sigma)). Crude extracts were then centrifuge at 20,000×*g* for 30 min at 4 °C. The supernatant was passed through a Miracloth (Calbiochem) and used for each immunoprecipitation as an input. The input was incubated with 40 μL anti-HA matrix (Roche #11815016001) for 3 h at 4 °C. Immunocomplexes were washed four times with the extraction buffer without the protease inhibitor cocktail. After washing, the matrix was resuspended into 50 μL 2× SDS sample buffer for 5 min at 95 °C. The protein samples were separated by 12.5% SDS-PAGE gel (ATTO) and blotted onto an Immobilon-P membrane (Millipore), and immunodetected using 1:10,000 diluted anti-GFP (in this study) or anti-HA (Roche #11583816001) antibodies, followed by a 1:50,000 diluted anti-Rabbit HRP-conjugate (Pierce #31460) or anti-Mouse HRP-conjugate (Pierce #31430). The uncropped full-scan data of immunoblots and gel are shown in Supplementary Fig. 17.

**Affinity column purification of YFP-AHG1 complexes.** The three-week-old seedlings (10–20 gFW) grown on MS plates were incubated for 2 h in water (-ABA samples) before being treated with 100 μM ABA for 24 h (+ABA samples). Samples were ground to a powder in liquid nitrogen and resuspended into 2× times extraction buffer (50 mM Na-phosphate pH 7.4, 150 mM NaCl, 0.1% NP-40, 1 mM DTT and 1× protease inhibitor cocktail (Sigma)). Crude extracts were then centrifuged at 20,000×*g* for 30 min at 4 °C. The complete supernatant was passed through a Miracloth (Calbiochem) and 0.45 μm syringe filter (Starlab Scientific), and loaded onto an anti-GFP (in this study) conjugated 1 mL HiTrap NHS-activated HP column (GE Healthcare). Anti-GFP affinity columns were generated according to the manufacturer's instructions (GE Healthcare). After a 50 to 100 mL wash with the extraction buffer without the protease inhibitor cocktail, YFP control and complexed AHG1-interacting proteins were eluted into 10 mL elution buffer (0.3 M Glycine-HCl pH 3.0, 1 mM DTT and 1× protease inhibitor cocktail (Sigma)), and were fractionated each (0.5 mL per fraction), and were TCA precipitated. Western blot was performed using a 1:10,000 diluted anti-GFP antibody (in this study), followed by a 1:50,000 diluted anti-Rabbit HRP-conjugate (Pierce #31460). Gel staining was performed using Oriole fluorescent gel stain (Bio-Rad). The uncropped full-scan data of immunoblots and gel are shown in Supplementary Fig. 17.

**Mass spectrometry and data analysis.** Purified proteins were reduced with 5 mM Tris (2-carboxyethyl) phosphine hydrochloride (Sigma-Aldrich) and alkylated. Proteins were digested for 18 h at 37 °C in 2 M urea, 100 mM Tris pH 8.5, 1 mM CaCl$_2$ with 2 μg trypsin (Promega). Analysis was performed using an Eksigent nanopump and a Thermo LTQ Orbitrap using an in-house built electrospray stage[68].

Protein and peptide identification and protein quantitation were done with Integrated Proteomics Pipeline—IP2 (Integrated Proteomics Applications, Inc., San Diego, CA. http://www.integratedproteomics.com/). Tandem mass spectra were extracted from raw files using RawConverter[69] and were searched against the

TAIR10_pep_20101214 database (https://www.arabidopsis.org/index.jsp) with YFP-AHG1, YFP, and reversed sequences added using ProLuCID[70,71]. The search space included all full-tryptic and half-tryptic peptide candidates for the tryptic digest with static modification of 57.02146 on cysteine. Peptide candidates were filtered using DTASelect[72], with these parameters: -p 2 -y 1 --trypstat --pfp 0.01 --extra --pI -DM 10 --DB --dm -in -t 1 --brief --quiet[69,72].

**Double and triple mutants and double-expression lines**. The *ahg1-1ahg3-1* double mutant was crossed with *hai1-2*, *aip1-1*, and *hai3-1*. Triple mutants were obtained from F2 or F3 progeny using mutant-specific CAPS markers and primer sets (Supplementary Table 4). The *ahg1-1* mutant was crossed with *dog1-2* and *dog1-3*. Double mutants were obtained from F2 or F3 progeny using mutant-specific CAPS markers and primer sets (Supplementary Table 4). HA-AHG1ox line was crossed with YFP-DOG1ox #2 line. The double-expression homo line was screened for hygromycin and bialaphos resistance, and obtained from homozygous T3 lines.

**RNA isolation and RT-PCR experiments**. Total RNA was isolated with a RNeasy Plant mini kit (Qiagen). After treatment with RNase-free DNase I (Qiagen), first-strand cDNA was synthesized from 1 μg total RNA using Superscript III first-strand synthesis super mix for qRT-PCR (Invitrogen), according to the manu-facturer's instructions. DNA fragments for *AHG1*, *DOG1* and *18s rRNA* were amplified for 20 or 30 PCR cycles by using gene specific primers (Supplementary Table 4). The uncropped full-scan data of gel are shown in Supplementary Fig. 18.

**Cloning for protein expression**. Full-length *DOG1* was cloned into the *Nde*I/*Bam*HI sites of the expression vector pCold I (TaKaRa) and pET30b (+) (Nova-gen) to obtain the N-terminal His₆-tagged and untagged DOG1, respectively. Full-length *DOGL3* and *DOGL5.2* cDNAs were inserted between *Nde*I/*Xho*I and *Nde*I/*Bam*HI sites of the pET28b (+) (Novagen), respectively, containing an N-terminal His₆-tag. The catalytic PP2C domain of *AHG1* (residues 104–416) cDNA was cloned in the pET28b (+) using *Nde*I/*Bam*HI sites with In-Fusion system (Clon-tech). Site-directed mutagenesis was performed with the QuickChange mutagenesis protocol (Stratagene). All plasmid constructs were confirmed by DNA sequencing and were introduced in the *Escherichia coli* BL21 (DE3) strain (Merck).

**Protein expression and purification**. To obtain apo- or heme-bound DOG1 expression, *E. coli* cells carrying expression plasmid were grown at 37 °C to reach an OD₆₀₀ of 0.5–0.6 in LB medium or TB medium, respectively, each containing antibiotics. Apo-DOG1 was induced by addition of IPTG to 0.2 mM and cell growth was continued for 16 h at 16 °C in LB medium (condition I). Heme-bound DOG1 was induced with 0.1 mM IPTG at 16 °C for 50–60 h in TB medium (condition II). The harvested cells expressing His₆-tagged DOG1 were resuspended in buffer A (20 mM Tris-HCl pH 8.5, 150 mM NaCl, 1 mM DTT, and 10% gly-cerol). The cells were lysed by sonication, and the debris was removed by cen-trifugation at 40,000×g for 30 min. The supernatant was purified with a 5 ml HisTrap HP column (GE Healthcare) and was further purified by HiLoad 26/60 Superdex 75 prep grade (GE Healthcare) pre-equilibrated with buffer B (10 mM Tris-HCl pH 8.5, 150 mM NaCl, and 5 mM DTT). DOGL3, DOGL5.2, and DOG1 mutants were prepared in the same manner. The untagged DOG1-expressed cells were resuspended in buffer C (20 mM Tris-HCl pH 8.5, 5 mM DTT) and lysed by sonication. Following centrifugation clarification, the supernatant was loaded to HiLoad 16/10 Q sepharose HP column (GE Healthcare) equilibrated with buffer C, and eluted with linear gradient from 0 to 500 mM NaCl. Fractions containing the untagged DOG1 were applied to a HiLoad 26/60 Superdex 75 prep grade (GE Healthcare) pre-equilibrated with buffer B. Further purification was performed by Mono Q 10/100 GL (GE Healthcare) equilibrated with buffer C. The untagged DOG1 was eluted in a linear gradient from 0 to 250 mM NaCl. The catalytic PP2C domain of AHG1 was expressed and purified by employing the same procedures used for the apo-DOG1. The purity of the expressed proteins was confirmed by 15% SDS-PAGE.

**PP2C enzyme assay**. Phosphatase activity of AHG1 was measured in 50 μL reaction buffer containing 25 mM Tris-HCl (pH7.5), 10 mM MgCl₂, 1 mM DTT with or without 50 μM ABA using Serine/Threonine phosphatase assay system (Promega). The phosphopeptide HSQPK(pS)TVGPTP, corresponding to the reg-ulatory phosphorylation site of SnRK2s, was synthesized and purified by Scrum Inc. Using synthetic SnRK2s phosphopeptide as substrate, the reaction buffer was added, which is 100 μM SnRK2s phosphopeptide and 200 μM AHG1ₐ₎ (1-103) with or without 800 μM heme-bound DOG1. Using an artificial substrate for phos-phatase 2A, 2B, and 2C, reaction buffer was added, which is 25 μM RRA(pT)VA peptide (Promega) and 50 μM AHG1ₐ₎ (1-103) with or without 200 μM heme-bound DOG1. After 30 min at 30 °C, each reaction was stopped with 50 μL molybdate dye solution. The absobance at 600 nm was measured with a plate reader (GloMax-Multi + Detection system; Promega).

**UV–visible absorption spectroscopy**. Electronic absorption spectra of the pur-ified recombinant proteins were measured on a CARY 400 Bio ultraviolet-visible

spectrophotometer (VARIAN) between 250 and 800 nm at room temperature in buffer B. For the titration experiment, hemin stock solution was prepared by dissolving 4 mg hemin (Wako) in 500 μL of 0.1 M NaOH and then centrifuged at 20,000×g for 15 min. Concentration of the hemin stock solution was determined using an extinction coefficient value of 58,440 M⁻¹ cm⁻¹ at 385 nm[73]. The solution was diluted with 20 mM Tris-HCl buffer pH 8.5, containing 5 mM DTT just before use. The hemin titration experiments were carried out on the untagged and His₆-tagged DOG1, and its His₆-tagged H245A and H245AH249A mutants by stepwise addition of a 1 μL aliquot of 1 mM hemin solution to 1 mL of 10 μM apo-proteins in 20 mM Tris-HCl buffer pH 8.5, containing 5 mM DTT. Samples were incubated for 5 min at room temperature before measuring the absorption spectra. The stoichiometry and affinity of hemin binding to apo-DOG1 were determined by nonlinear curve fitting of an increase in the absorbance at 425 nm for the untagged and His₆-tagged DOG1 at 418 nm for the His₆-tagged DOG1^H245A and at 416 nm for the His₆-tagged DOG1^H245AH249A, as described in the Methods. To obtain the absorption spectra of the fully heme-bound wild-type DOG1 and its mutants, the corresponding proteins in their apo forms were supplemented with 50 μM hemin dissolved in DMSO and incubated for 20 min at room temperature before gel-filtration chromatography, which removed the unbound hemin. It is worth men-tioning that the fully heme-bound proteins prepared hemin in NaOH and in DMSO displayed essentially the same absorption spectra including peak positions and heights. Protein concentration was estimated, as described in the Methods.

**Dissociation constant analysis**. The dissociation constants, $K_d$, of heme to DOG1 and its H245A and H245AH249A mutants were determined by fitting the increase of absorption at 425 nm for DOG1, 418 nm for the H245A mutant, and 416 nm for the H245AH249A mutant using following equations:

$$A_{obs} = \varepsilon_{DH}[DH] + \varepsilon_{H}[H]_f,$$

$$[DH] = ([D]_{total} + x[H]_{total} + K_d - \{([D]_{total} + x[H]_{total} + K_d)^2 - 4x[H]_{total}[D]_{total}\}^{1/2})/2,$$

$$[H]_f = [H]_{total} - [DH] \text{ and } [D]_{total} = [DH] + [D]_f,$$

where [DH] represents the concentration of the heme-bound protein while $[D]_f$ and $[H]_f$ are concentrations of the free protein and free hemin, respectively. The extinction coefficients of the protein-bound heme, $\varepsilon_{DH}$, and the free hemin, $\varepsilon_H$, were treated as variable parameters. The "$x$" represents a fraction of the active hemin, which can be incorporated into DOG1 and its relatives more easily than the aggregated hemin. We treated this "$x$" as a variable parameter because it is difficult to estimate the exact value due to co-existence of the monomeric and aggregated forms of hemin in the solution. Typically, the obtained "$x$" is larger than 0.8 in our experiments.

**Protein concentration**. Since free heme molecules could be removed almost completely during the chromatographic purification processes, concentrations of chromatographically purified proteins, with the exception of the H245A and H245AH249A mutants of DOG1, were estimated from the absorbance at 280 and 425 nm using the following equations:

$$[D]_{total} = [DH] + [D]_f,$$

$$[DH] = A_{425}/\varepsilon_{DH}(425),$$

$$[D]_f = \{A_{280} - \varepsilon_{DH}(280)[DH]\}/\varepsilon_D(280),$$

where $A_{280}$ and $A_{425}$ are the observed absorbance at 280 and 425 nm. The values of $A_{418}$ and $A_{416}$ were used for the H245A and H245AH249A mutants of DOG1, respectively. For the other proteins, the $A_{425}$ values were used. The extinction coefficient of the apo-protein at 280 nm, $\varepsilon_D(280)$, was evaluated from the amino acid sequence of the protein. The extinction coefficient of the heme-bound DOG1 at 425 nm, $\varepsilon_{DH}(425)$, was obtained from the $K_d$ analysis of the hemin titration experiments, and then $\varepsilon_{DH}(280)$ was calculated dividing $\varepsilon_{DH}(425)$ by a factor of 1.43, the Reinheitszahl value ($A_{425}/A_{280}$) determined for the fully heme-bound DOG1. The $\varepsilon_{DH}(425)$ and $\varepsilon_{DH}(280)$ thus obtained were used for all the proteins. Concentrations of the chromatographically purified H245A and H245AH249A mutants were estimated in the same manner using $A_{418}$ and $A_{416}$, respectively, instead of $A_{425}$.

**Circular dichroism spectroscopy**. Circular dichroism (CD) spectra of the purified recombinant DOG1 were recorded on a J-720W spectropolarimeter (Jasco) in the far-UV range from 185 to 260 nm using a 0.2-mm thermostatted cell at 25 °C, using the following parameters: resolution, 0.2 nm; bandwidth, 1.0 nm; sensitivity, 50 mdeg; response, 1 s; speed, 100 nm/min; accumulation, 8. Solutions used to record CD spectra contained apo-DOG1 or heme-bound DOG1 dissolved in 10

mM sodium phosphate buffer pH 7.5, with 5 mM DTT at protein concentration of 20 μM.

**Data availability**. The mass spectrometry proteomics data have been deposited to the ProteomeXchange Consortium via the PRIDE[74] partner repository with the dataset identifier PXD009392. All relevant data that support the findings of this study are available from the corresponding author upon reasonable request.

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

## Acknowledgements

We are grateful to Dr. Claire Delahunty and to Mr. Nathaniel Duncan for critical reading and comments on the manuscript. We thank Drs. Wim Soppe and Kazumi Nakabayashi for dog1-2 seeds, Dr. Paul Verslues for hai1-2, aip1-1 and hai3-1 seeds, Dr. Manabu Nakayama for pACTGW-attR and pASGW-attR vectors, Dr. Taku Demura for pH35YG vector, Dr. Yasuo Niwa for sGFP vector, ABRC for providing plant materials or reagents, Dr. Mitsuhiro Miyazawa for measuring CD spectra, and Ms. Izumi Odano for technical assistance. This research was supported by MEXT/JSPS KAKENHI (Grant Numbers 23688044, 26712030, and 23119521) and MEXT as part of Joint Research Program implemented at the Institute of Plant Science and Resources, Okayama University in Japan (2608, 2707, and 2805) to N.N., MEXT KAKENHI (15H05956) to T.K., National Institutes of Health (GM060396) to J.I.S., the National Institute of General Medical Sciences (8 P41 GM103533) to J.J.M and J.R.Y., and the National Institute of Agrobiological Sciences Strategic Fund to T.Y.

## Author contributions

N.N., W.T., J.I.S., and T.Y. designed the research; N.N., W.T., J.J.M., Y.H., K.S., T.I., and N.K. performed the experiments; N.N., W.T., J.J.M., Y.H., K.S., T.K., J.I.S., J.R.Y., T.H., and T.Y. analyzed the data; N.N., W.T., J.J.M., J.I.S., T.H., and T.Y. wrote the manuscript.

## Additional information

**Competing interests:** The authors declare no competing interests.

