## [Peer Review File · Nature Communications]

Reviewers' comments:

Reviewer #1 (Remarks to the Author):

In this manuscript the authors report on the physical interaction of the PP2C phosphatase AHG1 with DOG1, which pivotal role in seed dormancy was described in numerous studies but function remains elusive. Except recent evidence of genetic interactions with ABI3, no connections between DOG1 and ABA signaling pathway have been described. Therefore these novel findings about direct interactions between DOG1 and clade A PP2Cs constitute an important advance in the field. Another novel interesting result in this study is heme binding to DOG1, which opens new hypotheses for its function.

The authors performed an in-depth study of AHG1 interactions, testing both physical interactions by biochemical approaches (Y-2H, co-purification, co-IP) and genetic interactions using mutants and overexpressing lines. Interestingly, they also show that, PP2Cs of ABI1 sub-clade interact with PYR1 but not with DOG1 and vice versa for AHG1 sub-clade (except AHG3). Then based on these observations, they suggest a working model of DOG1 and ABA signaling pathways.

Overall the results are clear, well presented and convincing. The only reservation concerns the too limited investigation of the impact of DOG1 binding on PP2C function and downstream SnRK2 activities, which further study would complete the work. Nonetheless, other valuable data are presented that widen the interest of the whole study, notably about DOG1 interacting domain, PP2C interactions with DOG-like genes and structural analysis of heme binding, which all constitute novel findings.

Specific comments:

- AHG1-interacting proteins were identified by co-purification with YFP-AHG1 from 3-week old overexpressing plants. It indicates that DOG1 expression and function might not be restricted to seeds, the authors may therefore comment on this.
- In figure legends showing germination data, the authors indicate that error bars show s.d. of three independent experiments. They should make clear whether it means independent biological replicates (seed batches from 3 independent cultures) or the same batch was tested three times.
- For germination assays it would have been preferable to use a water medium instead of MS, since nutrients such as nitrate are known to release dormancy.
- Figure 3: data are only shown for a single YFP-AHG1 overexpressing line. It should be indicated whether other independent lines were tested with similar results. Moreover in root elongation and stomatal aperture assays, only small differences between YFP and YFP-AHG1 lines were observed, therefore a statistical analysis should be included.
- Concerning, subcellular localization of YFP-proteins, the reproducibility of observations on independent transgenic lines should be indicated. Furthermore most of these figures are very small, often preventing convincing interpretation.

Reviewer #2 (Remarks to the Author):

Authors provide interesting and novel data about genetic and physical interaction between DOG1 and AHG1 and connect these relations to ABA response and seed dormancy. Also DOG1-related proteins (DOGL1-5) have been examined. Additionally hem binding to DOG1 was found to be required for DOG1 function in seed dormancy.

This paper will be of high interest to the readers.

Authors describe:

- 1) interaction of PYR1 with clade A PP2Cs using yeast two hybrid (Y2H) and CoIP experiments in *N. benthamiana* confirming interaction of PYR1-AHG3 but not PYR1-AHG1
- 2) examined ABA response (germination assays) and effect of stratification length of several clade A PP2C triple mutants identifying the *ahg1-ahg3-hai3* triple mutant as have the stronger dormancy.
- 3) creation and phenotypical analysis of plant overexpressing YFP-AHG1 (strong ABA-insensitive phenotype in seed germination, normal plant size)
- 4) Identification of DOG1 as AHG1 interacting protein (AHIP) using YFP-AHG1 plants for affinity co-purification followed by LC-MS-MS and confirmed DOG1-AHG1 interaction by Y2H and CoIP (using *N. benthamiana* protoplasts)
- 5) created DOG1ox plants and analysed ABA-dependent germination phenotypes
- 6) Y2H analysis between AHG1 and DOGL1-5 proteins showed interaction of DOGL3 and DOGL5.2 with AHG1 followed by creation and analysis (germination efficiency) of plants overexpressing YFP-DOGL3 and YFP-DOGL5.2
- 7) creation and analysis of *ahg1/dog1* double mutant plants (exhibiting reduced germination like *ahg1* plants) and plants overexpressing both HA-AHG1 and YFP-DOG1 (exhibiting strong ABA insensitivity similar to HA-AHG1 ox plants) leading to the conclusion that DOG1 functions upstream of AHG1 in an ABA-dependent manner and probably directly regulates phosphatase activity.
- 8) phosphatase assay using recombinant AHG1 with/without DOG1 +/- ABA showed that DOG1 did not affect phosphatase activity in vitro.
- 9) Identification of DOG1 N-terminal region responsible for interaction with AHG1 (DOG1 delta 1-18) and creation of transgenic plants overexpressing DOG1(delta1-18), this plants showed similar germination efficiency as control line leading to the conclusion that i) N-terminal domain of DOG1 is required for DOG1 function and implying that ii) interaction with AHG1 is important for DOG1 function. More detailed analysis of additional DOG1 deletion mutants revealed residues 13-18 (DSYLEW) being essential for interaction with AHG1.
- 10) hem protein binding to DOG1 was identified by electronic absorption spectroscopy and confirmed by in vitro titration experiment, circular dichroism (CD) analysis revealed no effect of hem binding to secondary structure of DOG1. H245 and H249 were identified as potential axial ligands for bound heme in DOG1
- 11) Binding of DOG1 to AHG1 seems to act independent of DOG1 binding to hem, however of hem-binding of DOG1, DOGL3 and DOGL5.2 seems to collocate with observed ABA-insensitivity of respective DOG overexpressor lines.
- 12) DOG1-H245A ox lines in the presence of ABA showed lower germination and post-germination growth efficiencies and support relevance of DOG1 hem binding function in ABA signaling.

The performed experiments are clearly described and illustrated.

Major point:

Related to model presented in Figure 9:

As the in vitro phosphatase assay did not show change in PP2C activity the authors should provide kinase assays using SnRK2s as AHG1 substrate (+/- DOG1) to validate the presented model.

Reviewer #3 (Remarks to the Author):

I found several serious drawbacks in this paper, regarding the role of heme in the function. There is a large gap between the spectroscopic studies of the heme binding and functions (including germination and post-germination growth). I don't see any direct evidence that the heme binding to DOG1 regulates those functions. Studies of effects of heme and effects of mutations at the putative heme binding sites on the protein-protein (DOG1 and AHG1) interaction need to be carried out, at least, in order demonstrate that heme binding to DOG1 really regulates those functions. In addition, spectral studies are inconsistent with each other. The quality of this work does not meet the stringent standard of the Journal. Thus, I don't recommend publication of this work in Nature Communications.

Concerns that need to be addressed.

Page 11, line 185: The authors confirmed the specific interaction between DOG1 and AHG1. The authors need to examine if this interaction occurs in the presence or absence of heme.

Fig. 8c: The authors claim that H245AH249A is ferric high spin around 630 nm and the solution has a mixture of two species. But, to my eyes, this double mutant lost heme binding ability as demonstrated in Supp. Fig. 11c. The double mutant in Fig. 11c has no colour, probably due to the precipitates of the free hemin, since the hemin has very low solubility in water. I imagine that the absorption of the double mutant in Fig. 8c would be that of free hemin, because the band shape of the absorption is very similar to the free hemin. But, the absorption intensity of the free hemin should be significantly lower (since it takes dimer or aggregates in water) than that of the hemin bound to the protein, while the spectrum of Fig. 8c is not so. Did the authors purposely intensity the absorbance of the double mutant in Fig. 8c? Or does the hemin bind to the protein in nonspecific way? Then, why is the double mutant solution colourless (Supp. Fig. 11c)?

Fig. 8d: I don't see any difference in post-germination between the WT and double mutant. But, Supp. Fig. 12d shows the single mutation H245A significantly reduces the growth. I don't understand why the single mutant having small amount of heme significantly affect the function, whereas the double mutant, which lost heme binding ability, did not affect the function.

Again, the same as Fig. 8c as described above. Page 17, line 291: The authors say "DOG1 H245A shows faint red colour, while DOG1 H245AH249A is colourless" (Supp Fig. 11c). However, Fig. 8c shows both mutants have sufficient absorptions ascribed to heme with high intensities. All spectra in Fig. 8 indicate that all mutants have spectra with high intensity of heme so as to have red colour. Thus, results are inconsistent.

Supp. Fig. 5: H245 is not conserved, whereas H249 appears well conserved. But, the authors emphasize that H245 is more important for function.

Page 18, line 314: The paper describes "In the presence of ABA, YFP-DOG1H245AH249ox showed similar germination and post-germination----(Fig. 8d, Suppl. Fig. 13c), while YFP-DOG1H245Aox lines exhibited lower germination and---- (Supp. Fig. 12c,d). However, the authors demonstrate that the double mutant H245AH249A lost heme binding ability (Supp. Fig. 11c), although the spectrum of DOG1 H245AH249A has still heme bound to the protein, leading to inconsistency.

Supp.Fig.14-1, 14-2: No figure legends are attached to explain which column corresponds with which protein under various conditions. Even if the authors cite figure numbers, it would be beneficial for readers to incorporate legends here.

Again, page 21, line 376: Fig. 8c demonstrates that the double mutant H245AH249A has similar growth as WT. However, Supp. Fig. 12d shows that the single mutant H245A has strong inhibitory effect on the post-germination growth. Those results suggest that only single mutant H245A (partly heme-bound form) has strong effect, where the double mutant (lost heme-binding ability) has no effect. There is significant inconsistency. The authors already demonstrate that the single mutant H245A binds heme (small quantity), whereas the double mutant H245AH249A lost heme binding ability in Supp. Fig. 11c. Those results do not corroborate/prove that the heme binding to DOG1 is prerequisite for the functions.

The statement "Conversion of the histidine residue.....alanine abolish the DOG1 activity to confer....." is wrong in that the mutation certainly changed the heme coordination structure, but there is no direct evidence that this mutation abolished the activity.

I strongly urge the authors to examine the protein-protein interaction between DOG1 and AHG1 both in the presence or absence of hemin and between the wild type, the single mutant H245A, the double mutant H245AH249A of DOG1 and AHG1. Please do the same experiments of Figs.4 and 6 with/without heme and using the single and double mutants of DOG1. Otherwise, it is totally unclear how and at which point the heme binding to GOG1 influences the function. This paper lacks important biochemical and/or protein science studies that should fill the large gap and as well as demonstrate the role of heme in the biological functions.

I suspect that the heme-binding to DOG1 could influence interactions with other unknown targets described in Fig. 9. Then, the authors need to identify the targets. This paper is not warranted for publication in the Journal.

Reviewer #4 (Remarks to the Author):

The studies by Nishimura et al evaluate the control of seed dormancy by one of nine distinct PP2C enzymes, namely AHG1. More specifically, the studies analyzed the impact of either deleting or overexpressing AHG1 and related PP2C enzymes on selected aspects of ABA-dependent biology. The data as presented – discussed below - paint a somewhat confusing or unclear picture of how the AHG1/DOG1 complex functions to transduce ABA signals and regulate seed germination. Specifically, the lack of impact of PP2C phosphatase activity, albeit analyzed in vitro, fails to address the targets of phosphorylation or dephosphorylation that might control seed germination. Other aspects of work e.g. the proposed incorporation of Heme during the expression of DOG1 protein in E.coli – heme incorporation was not directly demonstrated in plants but inferred from mutations of two putative Heme-binding histidines– for 5 days, makes for a somewhat disjointed picture of how ABA signaling is transduced by AHG1 or DOG1.

1. The plant hormone, ABA, functions via the PYR1 receptor. Their initial studies (Figure 1) undertook a systematic or comprehensive analysis of ABA-dependent PYR1 binding to all nine members of the AHG and ABI families of PP2C enzymes using the yeast two-hybrid assay. These studies showed that PYR1 bound AHG3 and several members of the ABI family in the presence of ABA. More specifically, PYR1 failed to show any binding to AHG1 under the same conditions. The difference in PYR1 binding to AHG1 and AHG3 was also confirmed by IP experiments albeit the impact of ABA on AHG3 binding was much more modest. In any case, it is unclear why the authors considered the direct association with

the PYR1 receptor as a means to reveal how ABA communicates to AHG1, which in their hands, like many other AHG1 family members, was exclusively or "predominantly" nuclear. While one presumes that the PYR1 receptors possessed a quite distinct localization, perhaps plasma membrane or at least non-nuclear? So this figure offers no usual information in terms of explaining the subsequent studies.

2. Analysis of selected PP2C double and triple mutant lines (Figure 2) highlighted the significant impact of the loss of function of AHG1 and AHG3 on seed germination, that could be further enhanced by elimination of selected members of ABI1 family of PP2C enzymes e.g. HAI1 and HAI3 particularly at the earlier time ("0 days") but not at "4 days" but this information likely highlighting the non-redundant contributions of various PP2C enzymes in seed germination was not pursued further.

3. In assessing the impact of AHG1 on various aspects of ABA biology, the authors assessed the impact of overexpressing AHG1 on post-germination, root elongation and stomatal aperture (Figure 3) and found a selective impact in inhibiting ABA-mediated inhibition of post-germination growth. While this selectivity in ABA responses provided useful controls for subsequent analyses of AHG1 and binding partners, the lack of effect on either root elongation or stomatal aperture of either AHG1 or its binding partners was not analyzed.

4. Focusing on DOG1, identified among several proteins in IP-mass spec studies as a AHG1 binder, the authors showed it binding to multiple members of AHG1 family with only AHG1, AHG3 and HAI1 showing AGA dependency for this protein-protein interaction in the two-hybrid assay (Figure 4). For confirmation, the authors used IP experiments, which showed an easily discernable association of DOG1 with AHG1, albeit independent of ABA, the association with AHG3 was much weaker, given the higher levels of AHG3 expression in the input samples, than predicted by two-hybrid assay. Regardless, DOG1 overexpression, like AHG1 overexpression resulted in ABA insensitivity in post-germination growth.

5. To assess the interplay between AHG1 and DOG1 and structurally related proteins, the authors analyzed post-germination growth in mutant and overexpressing plants. While the ABA insensitivity elicited by mutation of AHG1 was not mimicked by DOG1 mutations or further enhanced by double mutations of AHG1 and DOG1, DOG1 overexpression had a negative impact on ABA-mediated post-germination which was greatly reduce by co-expression of AHG1 (Figure 5).

6. Figure 6 focuses on the structure-function analysis of DOG1, specifically focusing on domains required for AHG1 binding. However, several of the mutant DOG1 proteins failed to express and thus provided no useful information. In the limited few that were expressed, the deletion of N-terminal 18 amino acids, previously implicated in DOG1 dimerization were shown to be critical for AHG1 binding and for DOG1-mediated inhibition of post-germination growth. More detailed dissection of this region again yielded different levels of expression of the mutant DOG1 proteins and if anything highlighted the most significant binding of AHG1 to those DOG1 proteins that were the least well expressed (e.g. delta 1-6 delta 1-12 and delta 7-12). The authors failed to comment on this disparity.

7. Figures 7 and 8 focused on the putative heme-binding to DOG1 based primarily on the unique spectrum for DOG1 expressed in *E. coli* over 5 days. Mutation of putative heme-binding residues – H245 and H249 – that eliminated or perturbed the spectrum appeared to abrogate DOAG1 function in inhibiting post-germination growth and was inferred as a requirement for heme-binding to DOG1 for modulating seed germination but its role in AHG1 function was not assessed.

8. Together the data did not direct assess or support most aspects of the scheme or signaling pathway shown in Figure 9 and the value or validity of this proposed pathway will require a lot more experiments.

Detailed response to reviewers:

We thank the reviewers for their helpful comments. We have revised the manuscript and added new data according to the reviewers' comments and questions. We have addressed the reviewer comments point-by-point as described in the following.

Reviewer #1:

1. AHG1-interacting proteins were identified by co-purification with YFP-AHG1 from 3-week old overexpressing plants. It indicates that DOG1 expression and function might not be restricted to seeds, the authors may therefore comment on this.

Response: As requested by the reviewer, we have performed further experiments to assess DOG1 expression in three-week-old YFP-AHG1ox and YFPox control plants grown on MS plates and added the data in Supplementary Fig. 4d. These data show that the transcripts of *DOG1* were expressed in both YFP-AHG1ox and YFPox control plants. We have added text to the discussion to address this comment as well.

2. In figure legends showing germination data, the authors indicate that error bars show s.d. of three independent experiments. They should make clear whether it means independent biological replicates (seed batches from 3 independent cultures) or the same batch was tested three times.

Response: As requested by the reviewer, we have revised the text of each figure legends as follows: Error bars show s.d. of three independent experiments using the same seed batch.

3. For germination assays it would have been preferable to use a water medium instead of MS, since nutrients such as nitrate are known to release dormancy.

Response: We thank the reviewer for this suggestion. Since most of lines we tested showed an ABA hypersensitive phenotype at the seed germination stage, we think the effects of nutrients are minimal in our present data. In addition, many groups also have used MS medium for germination assays (Park et al., 2009, Antoni et al., 2012).

4. Figure 3: data are only shown for a single YFP-AHG1 overexpressing line. It should be indicated whether other independent lines were tested with similar results. Moreover in root elongation and stomatal aperture assays, only small differences between YFP and YFP-AHG1 lines were observed, therefore a statistical analysis should be included.

Response: We thank the reviewer for this suggestion. We found a few mistakes in Fig. 3c and have fixed them. As requested by the reviewer, we have performed root elongation and stomatal aperture assays using HA-AHG1 overexpression plants as a second line and added the data in Supplementary Fig. 3b,c. We also have performed a statistical analysis in root elongation and stomatal aperture assays and explained the information in each figure legend. These data show AHG1ox lines have ABA insensitive phenotypes in root growth but not stomatal responses. (Fig. 3c,d and Supplementary Fig. 3b,c).

5. Concerning, subcellular localization of YFP-proteins, the reproducibility of observations on independent transgenic lines should be indicated. Furthermore most of these figures are very small, often preventing convincing interpretation.

Response: As requested by the reviewer, we have changed the figure sizes in Fig. 3a and Supplementary Fig. 5b, 8b, 12b, 14b and 15b to show the subcellular localization of YFP-proteins on independent transgenic lines.

Reviewer #2:

1. Related to model presented in Figure 9:

As the *in vitro* phosphatase assay did not show change in PP2C activity the authors should provide kinase assays using SnRK2s as AHG1 substrate (+/- DOG1) to validate the presented model.

Response: We agree this experiment is important and could lead to a solid conclusion. Now we have performed *in vitro* phosphatase assays using a synthetic phosphopeptide corresponding to the regulatory phosphorylation site of SnRK2s and added the data in Fig. 5c. In our previous experiments using an artificial substrate for phosphatase 2A, 2B and 2C “RRA(pT)VA”, DOG1 could not inhibit the PP2C activity of AHG1 (Supplementary Fig. 10). The new data using a synthetic SnRK2 phosphopeptide show that DOG1 could reduce the PP2C activity of AHG1 (Fig. 5c). These data suggest that DOG1 regulates the activation state of SnRK2s through the inhibition of the PP2C activity of AHG1. Our presented model in Fig. 9 is not a conclusion but should be viewed as a new proposed model as the reviewers #3 and #4 also pointed out. We have revised Fig. 9 and the text to more carefully interpret the data presented in the manuscript.

Reviewer #3:

I found several serious drawbacks in this paper, regarding the role of heme in the function. There is a large gap between the spectroscopic studies of the heme binding and functions (including germination and post-germination growth). I don't see any direct evidence that the heme binding to DOG1 regulates those functions. Studies of effects of heme and effects of mutations at the putative heme binding sites on the protein-protein (DOG1 and AHG1) interaction need to be carried out, at least, in order demonstrate that heme binding to DOG1 really regulates those functions. In addition, spectral studies are inconsistent with each other. The quality of this work does not meet the stringent standard of the Journal. Thus, I don't recommend publication of this work in Nature Communications.

Response: We thank the reviewer for the comments. Since our manuscript might be unclear, the reviewer may have misunderstood relevant parts of our data. We reveal in this study that 1) DOG1 binds heme, 2) His245 and His249 play an important role in heme binding as an alternative axial ligand, 3) amino acid substitutions of these His residues significantly reduce heme binding ability, 4) overexpression of the DOG1 (H245A, H249A) cannot confer ABA hypersensitivity in post-germination growth, whereas wild type over-expression of DOG1 does. These data point to the model that heme binding of DOG1 is required for DOG1

function. We have revised the manuscript to make it more clearly understandable in the main text.

1. Page 11, line 185: The authors confirmed the specific interaction between DOG1 and AHG1. The authors need to examine if this interaction occurs in the presence or absence of heme.

Response: As requested by the reviewer, we have performed Co-IP experiments to assess the specific interaction between DOG1 and AHG1 with or without heme. We have added new data in Supplementary Fig. 13e showing that HA-AHG1 co-immunoprecipitates with YFP-DOG1 regardless of heme.

2. Fig. 8c: The authors claim that H245AH249A is ferric high spin around 630 nm and the solution has a mixture of two species. But, to my eyes, this double mutant lost heme binding ability as demonstrated in Supp. Fig. 11c. The double mutant in Fig. 11c has no colour, probably due to the precipitates of the free hemin, since the hemin has very low solubility in water. I imagine that the absorption of the double mutant in Fig. 8c would be that of free hemin, because the band shape of the absorption is very similar to the free hemin. But, the absorption intensity of the free hemin should be significantly lower (since it takes dimer or aggregates in water) than that of the hemin bound to the protein, while the spectrum of Fig. 8c is not so. Did the authors purposely intensity the absorbance of the double mutant in Fig. 8c? Or does the hemin bind to the protein in nonspecific way? Then, why is the double mutant solution colourless (Supp. Fig. 11c)?

Response: The protein samples used for Fig. 8c and Supplementary Fig. 11c (corresponding to Supplementary Fig. 13c in the revised manuscript) are different from one another. The samples used for Fig. 8c were prepared by incubating the colorless apo-DOG1 mutants with the excess amount of hemin followed by a gel filtration column chromatography to remove unbound hemin. This treatment produced the fully heme-bound DOG1^{H245A} and DOG1^{H245AH249A} with reddish brown color as much as the fully heme-bound wild-type DOG1. On the other hand, the recombinant wild-type and mutant DOG1 proteins expressed in *E coli* under condition II were used for the Supplementary Fig. 11c (corresponding to Supplementary Fig. 13c in the revised manuscript). Now we have added electronic absorption spectra of protein samples used for Supplementary Fig. 11c (corresponding to Supplementary Fig. 13c in the revised manuscript) in Supplementary Fig. 13d. Comparison of the spectra clearly demonstrates that the contents of the protein-bound heme are largely different between the protein samples used for Fig. 8c and Supplementary Fig. 13d in the revised manuscript. Proportions of the heme-bound forms in the protein samples used for Supplementary Fig. 13d in the revised manuscript were estimated to be 8.2% for DOG1^{H245A} and 3.3% for DOG1^{H245AH249A}. These values are significantly lower than the value of the wild-type DOG1 expressed under the same condition II (> 90%). These results suggest that both DOG1^{H245A} and DOG1^{H245AH249A} are able to bind heme (Fig. 8) but their heme-binding affinities are significantly lower than that of the wild-type DOG1. Therefore, we believe that H245 and H249 are critical for high affinity binding to heme but other residues also play important roles

in heme binding. In addition, we have now added the absorption spectrum of the free hemin in Fig. 8c, which is different from that of the fully heme-bound DOG1^{H245AH249A}. To make these points clearer, we have revised the manuscript and Fig. 8c as well as Supplementary Fig. 11 (corresponding to Supplementary Fig. 13 in the revised manuscript).

3. Fig. 8d: I don't see any difference in post-germination between the WT and double mutant. But, Supp. Fig. 12d shows the single mutation H245A significantly reduces the growth. I don't understand why the single mutant having small amount of heme significantly affect the function, whereas the double mutant, which lost heme binding ability, did not affect the function.

Response: YFP-DOG1ox lines have a lower post-germination growth efficiency than YFP control plants (Fig. 4c). YFP-DOG1^{H245A}ox lines have lower post-germination growth efficiency similarly to YFP-DOG1ox lines compared to YFP control plants (Supplementary Fig. 14d). These results suggested that DOG1^{H245A} still has DOG1 function *in vivo* to some extent, even if DOG1^{H245A} reduces the binding of heme compared to wild-type DOG1. In contrast, YFP-DOG1^{H245AH249A}ox lines show post-germination growth efficiency similarly to YFP controls (Fig. 8d), demonstrating that double mutation of H245 and H249 with alanine further reduced heme-binding ability and abolished *in vivo* DOG1 function almost completely. These data are consistent with the *in vitro* spectroscopic results (Fig. 8 and Supplementary Fig. 13).

4. Again, the same as Fig. 8c as described above. Page 17, line 291: The authors say “DOG1 H245A shows faint red colour, while DOG1 H245AH249A is colourless” (Supp Fig. 11c). However, Fig. 8c shows both mutants have sufficient absorptions ascribed to heme with high intensities. All spectra in Fig. 8 indicate that all mutants have spectra with high intensity of heme so as to have red colour. Thus, results are inconsistent.

Response: The same replies as those described in the response to the question 2. All the absorption spectra shown in Fig. 8 were measured for the fully heme-bound wild-type and mutant DOG1 proteins prepared by the treatment with the excess amount of hemin. That is why all the proteins show spectra with almost same intensity of the bound heme. On the other hand, absorption spectra of the protein samples used for Supplementary Fig. 11c (corresponding to Supplementary Fig. 13c in the revised manuscript) indicated that both DOG1^{H245A} and DOG1^{H245AH249A} showed significantly lower absorptions ascribed to the bound heme than wild-type DOG1 (Supplementary Fig. 13d). Our data are consistent.

5. Supp. Fig. 5: H245 is not conserved, whereas H249 appears well conserved. But, the authors emphasize that H245 is more important for function.

Response: We do not emphasize that H245 is more important for DOG1 function than H249. We report that both H245 and H249 are important for the DOG1 function based on our findings reported in the manuscript. Among DOG1/DOGL proteins shown in Supplementary Fig. 5 (corresponding to Supplementary Fig. 6 in the revised manuscript), only two proteins, DOG1 and DOGL3, showed both the heme-binding ability *in vitro* and the strong ABA

hypersensitive phenotype in seed germination *in vivo*. These two proteins have both H245 and H249. Conversion of H245 to alanine led to YFP-DOG1^{H245A}ox lines which have lower germination and post-germination growth efficiencies similarly to YFP-DOG1ox line than YFP-control line. Further conversion of H249 to alanine led to YFP-DOG1^{H245AH249A}ox lines which have lost DOG1 function almost completely. In addition, heme-coordination patterns of DOG1^{H245A} and DOG1^{H245AH249A} are different from that of the wild-type DOG1 due to the lack of the axial ligand for the bound heme. Hence, we conclude that both H245 and H249 are important for the DOG1 function.

6. Page 18, line 314: The paper describes “In the presence of ABA, YFP-DOG1H245AH249ox showed similar germination and post-germination----(Fig. 8d, Suppl. Fig. 13c), while YFP-DOC1H245Aox lines exhibited lower germination and---- (Suppl. Fig. 12c,d). However, the authors demonstrate that the double mutant H245AH249A lost heme binding ability (Suppl. Fig. 11c), although the spectrum of DOG1 H245AH249A has still heme bound to the protein, leading to inconsistency.

Response: As mentioned above in the response to the question 2, both DOG1^{H245A} and DOG1^{H245AH249A} are able to bind heme (Fig. 8) although their heme-binding affinities are significantly lower than those of the wild-type DOG1 (Supplementary Fig. 13c,d). The order of the heme-binding affinities is wild-type DOG1 > DOG1^{H245A} > DOG1^{H245AH249A}. And the order of the inhibition of germination and post-germination efficiencies is YFP-DOG1ox line > YFP-DOG1^{H245A}ox line > YFP-DOG1^{H245AH249A}ox line as shown in Fig. 8d and Supplementary Fig. 14c,d and 15c. Hence, the *in vivo* results correlate well with the *in vitro* spectroscopic results.

7. Supp.Fig.14-1, 14-2: No figure legends are attached to explain which column corresponds with which protein under various conditions. Even if the authors cite figure numbers, it would be beneficial for readers to incorporate legends here.

Response: According to the manuscript checklist on *Nature Communications*, we just included the full blot and gel picture where portions of blots and gels have been presented in the main paper. As requested by the reviewer, we have added some information to each figure which should be beneficial for readers.

8. Again, page 21, line 376: Fig. 8c demonstrates that the double mutant H245AH249A has similar growth as WT. However, Supp. Fig. 12d shows that the single mutant H245A has strong inhibitory effect on the post-germination growth. Those results suggest that only single mutant H245A (partly heme-bound form) has strong effect, where the double mutant (lost heme-binding ability) has no effect. There is significant inconsistency. The authors already demonstrate that the single mutant H245A binds heme (small quantity), whereas the double mutant H245AH249A lost heme binding ability in Supp. Fig. 11c. Those results do not corroborate/prove that the heme binding to DOG1 is prerequisite for the functions.

Response: We think our replies described in the response to the questions 2 to 6 would be the answer to this comment. Our data are consistent with between the *in vivo* seed germination results and the *in vitro* spectroscopic results.

9. The statement “Conversion of the histidine residue.....alanine abolish the DOG1 activity to confer.....” is wrong in that the mutation certainly changed the heme coordination structure, but there is no direct evidence that this mutation abolished the activity.

Response: We think our replies described in the response to the questions 2 to 6 would be the answer to this comment. We have revised and added text to render this subject more clearly understandable in the manuscript.

10. I strongly urge the authors to examine the protein-protein interaction between DOG1 and AHG1 both in the presence or absence of hemin and between the wild type, the single mutant H245A, the double mutant H245AH249A of DOG1 and AHG1. Please do the same experiments of Figs.4 and 6 with/without heme and using the single and double mutants of DOG1. Otherwise, it is totally unclear how and at which point the heme binding to GOG1 influences the function. This paper lacks important biochemical and/or protein science studies that should fill the large gap and as well as demonstrate the role of heme in the biological functions.

Response: As requested by the reviewer, we have performed Co-IP experiment to assess the specific interaction between AHG1 with both DOG1 single H245A and double H245AH249A mutations and these data are now added in Supplementary Fig. 13e. The new data show that HA-AHG1 co-immunoprecipitates with YFP- DOG1, YFP- DOG1^{H245A} and YFP- DOG1^{H245AH249A} regardless of heme. Our present data indicate that heme binding of DOG1 is important for DOG1 function but not for the interaction with AHG1. This statement is further supported by our new results of *in vitro* phosphatase assays using a synthetic phosphopeptide corresponding to the regulatory phosphorylation site of SnRK2s (Fig. 5c).

11. I suspect that the heme-binding to DOG1 could influence interactions with other unknown targets described in Fig. 9. Then, the authors need to identify the targets. This paper is not warranted for publication in the Journal.

Response: Our presented model in Fig. 9 should not be viewed as a final conclusion but points to a novel understanding that will most likely trigger many additional important investigations as reviewers #2 and #4 also pointed out. We have revised Fig. 9 and the text to more carefully interpret the data presented in the manuscript.

Reviewer #4:

The studies by Nishimura et al evaluate the control of seed dormancy by one of nine distinct PP2C enzymes, namely AHG1. More specifically, the studies analyzed the impact of either deleting or overexpressing AHG1 and related PP2C enzymes on selected aspects of ABA-dependent biology. The data as presented – discussed below - paint a somewhat

confusing or unclear picture of how the AHG1/DOG1 complex functions to transduce ABA signals and regulate seed germination. Specifically, the lack of impact of PP2C phosphatase activity, albeit analyzed in vitro, fails to address the targets of phosphorylation or dephosphorylation that might control seed germination. Other aspects of work e.g. the proposed incorporation of Heme during the expression of DOG1 protein in E.coli – heme incorporation was not directly demonstrated in plants but inferred from mutations of two putative Heme-binding histidines– for 5 days, makes for a somewhat disjointed picture of how ABA signaling is transduced by AHG1 or DOG1.

Response: We thank the reviewer for the comments. We have added new data and information and revised the manuscript to assess a confusing or unclear picture the reviewer pointed out.

1. The plant hormone, ABA, functions via the PYR1 receptor. Their initial studies (Figure 1) undertook a systematic or comprehensive analysis of ABA-dependent PYR1 binding to all nine members of the AHG and ABI families of PP2C enzymes using the yeast two-hybrid assay. These studies showed that PYR1 bound AHG3 and several members of the ABI family in the presence of ABA. More specifically, PYR1 failed to show any binding to AHG1 under the same conditions. The difference in PYR1 binding to AHG1 and AHG3 was also confirmed by IP experiments albeit the impact of ABA on AHG3 binding was much more modest. In any case, it is unclear why the authors considered the direct association with the PYR1 receptor as a means to reveal how ABA communicates to AHG1, which in their hands, like many other AHG1 family members, was exclusively or “predominantly” nuclear. While one presumes that the PYR1 receptors possessed a quite distinct localization, perhaps plasma membrane or at least non-nuclear? So this figure offers no usual information in terms of explaining the subsequent studies.

Response: As pointed out by the reviewer, we have added and revised the text of the results section. Subcellular localization experiments of PYR1 and PYLs have been already reported, showing that these are localized to both nucleus and cytosol (Park et al., 2009, Ma et al., 2009, Santiago et al., 2009). We previously identified AHG1 as a central negative regulator of ABA responses in seeds (Nishimura et al., 2007). *ahg1* mutant shows a strong ABA hypersensitive phenotype in germination and the levels of *AHG1* mRNA are up-regulated by ABA. Interestingly, we find that PYR1 interacts with AHG3 but not AHG1 (Fig. 1), even though these are predominantly localized in nucleus and classified into the same AHG1 subfamily. These data led us to offer the motivation of our present research to understand whether AHG1 has unique regulatory and functional mechanisms in ABA signaling at the seed dormancy and germination stages in parallel with PYL/RCAR receptor-dependent regulation or not.

2. Analysis of selected PP2C double and triple mutant lines (Figure 2) highlighted the significant impact of the loss of function of AHG1 and AHG3 on seed germination, that could be further enhanced by elimination of selected members of ABI1 family of PP2C enzymes e.g. HAI1 and HAI3 particularly at the earlier time (“0 days”) but not at “4 days” but this

information likely highlighting the non-redundant contributions of various PP2C enzymes in seed germination was not pursued further.

Response: We thank the reviewer for this comment. These data point to the model that at least AHG1, AHG3 and HAI3 in the AHG1 subfamily have overlapping but distinct functions. Our data show that these triple mutant seeds have a deeper dormancy, and further suggest that at least AHG1, AHG3 and HAI3 in the AHG1 subfamily of PP2Cs have an important function in the regulation of seed dormancy. We have added text to the results section to clarify this point.

3. In assessing the impact of AHG1 on various aspects of ABA biology, the authors assessed the impact of overexpressing AHG1 on post-germination, root elongation and stomatal aperture (Figure 3) and found a selective impact in inhibiting ABA-mediated inhibition of post-germination growth. While this selectivity in ABA responses provided useful controls for subsequent analyses of AHG1 and binding partners, the lack of effect on either root elongation or stomatal aperture of either AHG1 or its binding partners was not analyzed.

Response: As the reviewer requested, we have performed root elongation and stomatal aperture assays using HA-AHG1 overexpression plants as a second line and added the data in Supplementary Fig. 3b,c. These data show that AHG1ox lines exhibit an ABA insensitive phenotype at post-germination and root growth but not stomatal responses. We also performed root elongation and stomatal aperture assays using YFP-DOG1 overexpression plants and added the data in Supplementary Fig. 5d,e. The new data show that the inhibitory effect of ABA on root growth and stomatal responses of the YFP-DOG1ox lines were similar to the responses of the control plants, suggesting that AHG1 and DOG1 have overlapping but distinct physiological functions.

4. Focusing on DOG1, identified among several proteins in IP-mass spec studies as a AHG1 binder, the authors showed it binding to multiple members of AHG1 family with only AHG1, AHG3 and HAI1 showing AGA dependency for this protein-protein interaction in the two-hybrid assay (Figure 4). For confirmation, the authors used IP experiments, which showed an easily discernable association of DOG1 with AHG1, albeit independent of ABA, the association with AHG3 was much weaker, given the higher levels of AHG3 expression in the input samples, than predicted by two-hybrid assay. Regardless, DOG1 overexpression, like AHG1 overexpression resulted in ABA insensitivity in post-germination growth.

Response: As pointed out by the reviewer, we present that DOG1 interacts with all AHG1 subfamily members in yeast two-hybrid assays (Fig.4) but our yeast two-hybrid data do not show any ABA dependency in the interactions. And our data indicate that YFP-AHG3 bound less effectively to HA-DOG1 in tobacco leaves as the reviewer mentioned. AHG1ox lines showed strong ABA insensitivity at the post-germination stage, while DOG1ox lines show strong ABA hypersensitivity at the post-germination stage.

5. To assess the interplay between AHG1 and DOG1 and structurally related proteins, the authors analyzed post-germination growth in mutant and overexpressing plants. While the

ABA insensitivity elicited by mutation of AHG1 was not mimicked by DOG1 mutations or further enhanced by double mutations of AHG1 and DOG1, DOG1 overexpression had a negative impact on ABA-mediated post-germination which was greatly reduced by co-expression of AHG1 (Figure 5).

Response: The loss of function mutation of AHG1, such as *ahg1-1* showed an ABA hypersensitivity in post-germination growth while *dog1* mutants showed less post-germination growth independent of ABA. These data are consistent with previous reports (Bentsink et al., 2006, Nishimura et al., 2007). As pointed out by the reviewer, *ahg1dog1* double mutants showed an ABA hypersensitivity in post-germination growth like the *ahg1-1* mutant. Correspondingly, the DOG1/AHG1 double overexpression line showed an ABA insensitivity in the post-germination growth like the AHG1 overexpression line, while the DOG1 overexpression line showed an ABA hypersensitivity in post-germination growth. These results suggest that DOG1 acts upstream of AHG1.

6. Figure 6 focuses on the structure-function analysis of DOG1, specifically focusing on domains required for AHG1 binding. However, several of the mutant DOG1 proteins failed to express and thus provided no useful information. In the limited few that were expressed, the deletion of N-terminal 18 amino acids, previously implicated in DOG1 dimerization were shown to be critical for AHG1 binding and for DOG1-mediated inhibition of post-germination growth. More detailed dissection of this region again yielded different levels of expression of the mutant DOG1 proteins and if anything highlighted the most significant binding of AHG1 to those DOG1 proteins that were the least well expressed (e.g. delta 1-6 delta 1-12 and delta 7-12). The authors failed to comment on this disparity.

Response: As the reviewer requested, we have moved the data in Fig. 6a,b to Supplementary Fig. 11. We have tried to control the expression levels of target proteins in our Co-IP system as much as possible. Our detailed dissection of N-terminal 18 amino acids showed that HA-AHG1 co-immunoprecipitated with YFP-DOG1 Δ 1-6, YFP-DOG1 Δ 1-12 and YFP-DOG1 Δ 7-12, but not detectably or very weakly with YFP-DOG1 Δ 7-18 and YFP-DOG1 Δ 13-18 as well as YFP-DOG1 Δ 1-18. Thus we concluded that the six-residue sequence of DOG1 spanning position 13-18, DSYLEW, is essential for interacting with AHG1. We also have revised the text in the results as the reviewer requested. As the reviewer pointed out, we noticed that the reduction of HA-AHG1 levels when co-expressed with deleted forms of YFP-DOG1 that can interact with HA-AHG1 compared with deleted forms of YFP-DOG1 that cannot interact with HA-AHG1. Although we could not exclude the possibility that this is caused by experimental conditions, DOG1-interaction might affect AHG1 level.

7. Figures 7 and 8 focused on the putative heme-binding to DOG1 based primarily on the unique spectrum for DOG1 expressed in E. coli over 5 days. Mutation of putative heme-binding residues – H245 and H249 – that eliminated or perturbed the spectrum appeared to abrogate DOG1 function in inhibiting post-germination growth and was inferred as a requirement for heme-binding to DOG1 for modulating seed germination but its role in AHG1 function was not assessed.

Response: We expressed recombinant DOG1 in *E.coli* at 16 °C for 2-2.5 days (50-60h) not over 5 days in TB medium (condition II). We presented that DOG1 has the ability to bind heme and to AHG1, and that these binding abilities are independent process each other *in vivo*. As the reviewer requested, we have performed further experiments and added new data in Fig. 5c and Supplementary Fig. 13e. We show that H245AH249A mutation in DOG1 results in significantly reduced heme binding activity and perturbed heme coordination, but did not abolish the interactions with AHG1 (supplementary Fig 13e). We also show that DOG1 reduces the PP2C activity of AHG1 in an *in vitro* assay using a synthetic phospho SnRK2s peptide (Fig. 5c). These data suggested that DOG1-AHG1 complex seems to control downstream components such as SnRK2s at seed dormancy and germination.

8. Together the data did not direct assess or support most aspects of the scheme or signaling pathway shown in Figure 9 and the value or validity of this proposed pathway will require a lot more experiments.

Response: Our presented model in Fig. 9 is not a conclusion but should be viewed as a new working model as reviewers #2 and #3 also pointed out. We have revised Fig. 9 and the text to more carefully interpret the data presented in the manuscript.

References

- Antoni, R. *et al.* Selective inhibition of clade A phosphatases type 2C by PYR/PYL/RCAR abscisic acid receptors. *Plant Physiol.* **158**, 970-980 (2012).
- Bentsink, L., Jowett, J., Hanhart, C.J. & Koornneef, M. Cloning of *DOG1*, a quantitative trait locus controlling seed dormancy in *Arabidopsis*. *Proc. Natl Acad. Sci. USA* **103**, 17042-17047 (2006).
- Ma, Y. *et al.* Regulators of PP2C phosphatase activity function as abscisic acid sensors. *Science* **324**, 1064-1068 (2009).
- Nishimura, N. *et al.* *ABA-Hypersensitive Germination1* encodes a protein phosphatase 2C, an essential component of abscisic acid signaling in *Arabidopsis* seed. *Plant J.* **50**, 935-949 (2007).
- Park, S.Y. *et al.* Abscisic acid inhibits type 2C protein phosphatases via the PYR/PYL family of START proteins. *Science* **324**, 1068-1071 (2009).
- Santiago, J. *et al.* Modulation of drought resistance by the abscisic acid receptor PYL5 through inhibition of clade A PP2Cs. *Plant J.* **60**, 575-588 (2009).

Reviewers' comments:

Reviewer #1 (Remarks to the Author):

In the revised manuscript and response letter to reviewers the authors took into account my specific comments. Nevertheless I have few additional remarks concerning answers to the comments 2, 3 and 5.

Comments 2 and 3: As now specified, germination results are shown for a single seed lot per genotype. Were similar results obtained with seeds from independent cultures? Indeed, assessing seed germination and especially seed dormancy on a single seed batch is not satisfactory since environmental conditions encountered by the mother plant have been clearly shown to affect seed germination characteristics. Furthermore the use a single seed lot does not allow to ascertain the reproducibility of small differences between genotypes, such as shown in Figure 2. The authors conclude from figure 2c, that ahg1-1 ahg3-1 hai3-1 seeds have a deeper dormancy because they do not fully germinate (80 %) after a 4-day stratification treatment. However there might be a proportion of non-viable/poorly filled seeds in this particular seed lot, unless 100% germination was obtained under optimal conditions. The use of MS medium would be expected to facilitate maximal germination. Considering the study of seed dormancy is far from complete and unless similar results were obtained with seeds from independent cultures, the authors should tone down these conclusions in page 9.

Comment 5: The quality of the figure 3a convincingly supports the localization of YFP-AHG1 protein in the nucleus. This is less clear in the supplementary figures, in which nuclei are not well visible. Specific organelle staining may help.

Reviewer #2 (Remarks to the Author):

The authors have provided a substantially improved version of the manuscript.

minor points:

please keep commonly used nomenclature:

DOG1 is abbreviation of "Delay of Germination 1" (but not Delay of Seed germination 1)

Page 3 and page 21: "in an in vitro assay" change to "in vitro"

Page 4: "spouting" change to "sprouting"

Page 13-14 - Figure 5c/Suppl. Figure 10 and corresponding text describing the in vitro assay: the Figure 5c/Suppl. Figure 10 is labeled "AHG1" and "AHG1+DOG1" implying that full length proteins have been used. However in this assay a N-terminally truncated AHG1 version was used which was described only later in the next chapter; similarly the text on bottom of page 13 does not mentioned usage of the truncated AHG1 version in the in vitro assay, please amend.

Reviewer #4 (Remarks to the Author):

The authors have made a sincere effort to address all the reviewers' comments and as a result, the revised manuscript is greatly improved.

Reviewer #5 (Remarks to the Author):

Per the editor's request, I focus my comments on the DOG1-heme interaction aspect of the manuscript. I find the findings that DOG1 binds heme and that this interaction is important for regulating seed dormancy are intriguing. However, there are a number of technical issues that complicates the interpretation of the results. I'd like to see the authors addressing these issues before deciding whether to recommend publication at Nature Communications.

Major issues:

1. Fig. 7a, how are the protein concentrations determined? If using A280, how are the extinction coefficients determined? Heme contributes to A280. This should be taken into consideration when trying to accurately determine the protein concentration. Because of this, the actual concentrations of heme-bound proteins are much lower. MicroBCA can be used to measure it experimentally.
2. I don't understand how the K_d of 95.8 nM for heme is derived from the titration experiments shown in Fig. 7d. The details are not provided and this seems impossible to achieve using the method described.
3. Why aren't the heme affinities for the mutants measured?
4. The conclusion that AHG1 interaction with DOG1 is independent of heme binding in vivo is not convincing because the H245A and H245A/H249A mutants still have decent heme affinity. It has not been determined whether they bind heme in plants.
5. The interpretation of DOG1 H245A and DOG1 H245A/H249A overexpression phenotypes is complicated by the lack of knowledge on the relationship between the overexpressed mutants and endogenous wild-type protein. In this regard, it also helps to know the oligomerization states of DOG1, DOGL3 and DOGL5.2.

Further notes on the DOG1-heme interaction:

6. It looks like that DOG1 and DOGL3 are expressed with N-terminal His6 tags and DOGL5.2 is expressed with a C-terminal His6 tag. These tags are not removed during purification. The presence of the His6 tags can complicate the in vitro heme binding studies as they can also bind heme. Conversely, imidazole commonly used in His affinity purification can interact with free heme and favor dissociation of heme from the protein during the elution and any incubation before imidazole is removed. Usually it is better to use apo proteins without His tags in binding assays, and to avoid using imidazole during purification (for example employ ion exchange chromatography methods).
7. Two methods were used to prepare hemin solutions when studying DOG1-heme interaction, namely dissolution in either 0.1 M NaOH and in DMSO. Note that hemin has a chloride group serving as a strong ligand. When dissolved in high concentrations of NaOH, chloride is replaced by OH⁻. This does not happen in DMSO. The two methods should be compared to be sure that no differences arise.
8. Fig. 7c,d, it would be helpful to the readers if the apoDOG1 protein concentration is explicitly given in the figure legend. By the way, line 582, "mM" perhaps should be "μM." To understand how heme absorbs light after the DOG1 protein has been saturated, it would be interesting to generate a difference spectrum between 20 μM and 12 μM heme (the latter is a rough estimate by visual inspection). Also, relevant to the point above, the contribution of heme to extinction coefficient at 280 nm may be extrapolated from the titration.
9. Fig. 7d, the extinction coefficient of DOG1-bound heme may be estimated from the slope. Visual estimate suggest a value of around 70 mM⁻¹cm⁻¹.
10. 5 mM DTT is included in the protein and hemin solutions. The effects of DTT on the absorption spectra should be evaluated. First, DTT may serve as an exogenous co-axial ligand. Second, DTT at 5 mM concentration could reduce heme (although slowly). I suggest recording electronic absorption spectra in the absence of DTT, as well as preparing hemin solutions without DTT in the titration experiments.

Minor comments on other parts of the manuscript:

11. There seems no strong need to define AHIPs since every one of them already has a name. This will reduce confusion.

12. The subcellular localization studies rely on overexpression of fusions with fluorescence proteins. Both overexpression and fusion with a large tag protein could result in differences with the native proteins and potential misleading conclusions.

13. Fig. 8d, is the value for YFP-DOG1 at 0.3 μ M ABA zero?

Detailed response to reviewers:

We thank the reviewers for their helpful comments. We have revised the manuscript and added new data according to the reviewers' comments and questions. We have addressed the reviewer comments point-by-point as described in the following. Discussed changes are highlighted in the manuscript text.

Reviewer #1:

In the revised manuscript and response letter to reviewers the authors took into account my specific comments. Nevertheless I have few additional remarks concerning answers to the comments 2, 3 and 5.

Comments 2 and 3: As now specified, germination results are shown for a single seed lot per genotype. Were similar results obtained with seeds from independent cultures? Indeed, assessing seed germination and especially seed dormancy on a single seed batch is not satisfactory since environmental conditions encountered by the mother plant have been clearly shown to affect seed germination characteristics. Furthermore the use a single seed lot does not allow to ascertain the reproducibility of small differences between genotypes, such as shown in Figure 2. The authors conclude from figure 2c, that ahg1-1 ahg3-1 hai3-1 seeds have a deeper dormancy because they do not fully germinate (80 %) after a 4-day stratification treatment. However there might be a proportion of non-viable/poorly filled seeds in this particular seed lot, unless 100% germination was obtained under optimal conditions. The use of MS medium would be expected to facilitate maximal germination. Considering the study of seed dormancy is far from complete and unless similar results were obtained with seeds from independent cultures, the authors should tone down these conclusions in page 9.

Response: We thank the reviewer for the comments. As pointed out by the reviewer, we have toned down these conclusions in page 9.

Comment 5: The quality of the figure 3a convincingly supports the localization of YFP-AHG1 protein in the nucleus. This is less clear in the supplementary figures, in which nuclei are not well visible. Specific organelle staining may help.

Response: We thank the reviewer for this suggestion. As requested by the reviewer, we have added the photos were Normarski and merged images in Supplementary Fig. 1b. Our presented data are also consistent with subcellular localizations of some GFP fused to group A PP2C members (Umezawa et al., 2009).

Reviewer #2:

The authors have provided a substantially improved version of the manuscript.

minor points:

please keep commonly used nomenclature:

DOG1 is abbreviation of "Delay of Germination 1" (but not Delay of Seed germination 1)

Response: We are sorry for making this mistake. We have revised it in the manuscript.

Page 3 and page 21: "in an in vitro assay" change to "in vitro"

Response: We thank the reviewer for this suggestion. We have revised it in the manuscript.

Page 4: “spouting” change to “sprouting”

Response: We are sorry for making this mistake. We have revised it in the manuscript.

Page 13-14 - Figure 5c/Suppl. Figure 10 and corresponding text describing the in vitro assay: the Figure 5c/Suppl. Figure 10 is labeled “AHG1” and “AHG1+DOG1” implying that full length proteins have been used. However in this assay a N-terminally truncated AHG1 version was used which was described only later in the next chapter; similarly the text on bottom of page 13 does not mention usage of the truncated AHG1 version in the in vitro assay, please amend.

Response: We thank the reviewer for the comments. As pointed out by the reviewer, we have revised the manuscript and the text of each figure legends and labels in Fig. 5c and Supplementary Fig. 10.

Reviewer #4:

The authors have made a sincere effort to address all the reviewers' comments and as a result, the revised manuscript is greatly improved.

Response: We thank the reviewer for their comments.

Reviewer #5:

Per the editor's request, I focus my comments on the DOG1-heme interaction aspect of the manuscript. I find the findings that DOG1 binds heme and that this interaction is important for regulating seed dormancy are intriguing. However, there are a number of technical issues that complicate the interpretation of the results. I'd like to see the authors addressing these issues before deciding whether to recommend publication at Nature Communications.

Major issues:

1. Fig. 7a, how are the protein concentrations determined? If using A280, how are the extinction coefficients determined? Heme contributes to A280. This should be taken into consideration when trying to accurately determine the protein concentration. Because of this, the actual concentrations of heme-bound proteins are much lower. MicroBCA can be used to measure it experimentally.

Response: We thank the review for the comments. In the revised manuscript, we have estimated protein concentrations using both A280 and A425 as described in Supplementary Methods. Since free heme molecules could be removed almost completely during our chromatographic purification process, essentially no contributions of the free heme to A280 and A425 were expected for the chromatographically purified proteins used for the electronic absorption measurements.

2. I don't understand how the K_d of 95.8 nM for heme is derived from the titration experiments shown in Fig. 7d. The details are not provided and this seems impossible to achieve using the method described.

Response: We are sorry for not providing the details of how to derive K_d s from the titration experiments. In the revised manuscript, we have provided the detailed and more precise procedure of the K_d analysis in Supplementary Methods. This analysis provides not only K_d

value but also the extinction coefficients of the DOG-bound heme and the free hemin at 425 nm as well as a fraction of the active, most likely monomeric hemin which can be incorporated into DOG1.

3. Why aren't the heme affinities for the mutants measured?

Response: According to the reviewer's suggestion, we now conducted hemin titration experiments with the H245A and H245AH249A mutants of DOG1 and included these results in the revised manuscript as Supplementary Fig. 13 and Supplementary Table 3.

4. The conclusion that AHG1 interaction with DOG1 is independent of heme binding *in vivo* is not convincing because the H245A and H245A/H249A mutants still have decent heme affinity. It has not been determined whether they bind heme in plants.

Response: We presented that DOG1 has the ability to bind AHG1 and heme. We show that H245A and H245A/H249A mutations in DOG1 result in reduced heme binding activity and perturbed heme coordination *in vitro* (Fig. 8a,c and Supplementary Fig. 14d in the revised manuscript), but did not abolish the interactions with AHG1 *in vivo* (Supplementary Fig. 14e in the revised manuscript). YFP-DOG1^{H245A}ox lines have lower post-germination growth efficiency similar to YFP-DOG1ox lines when compared to YFP control plants (Supplementary Fig. 15d in the revised manuscript). These results suggest that DOG1^{H245A} still has DOG1 function *in vivo* to some extent, even if DOG1^{H245A} reduces the binding of heme compared to wild-type DOG1. In contrast, YFP-DOG1^{H245AH249A}ox lines show post-germination growth efficiency similar to YFP controls (Fig. 8d), demonstrating that double mutation of H245A/H249A abolished *in vivo* DOG1 function almost completely. Our data and conclusion are consistent with the *in vitro* spectroscopic results (Fig. 8 and Supplementary Fig. 14 in the revised manuscript) and supported by *in vivo* data. The conclusion that AHG1 interaction with DOG1 is independent of heme binding is further supported by the results obtained for DOG1 Δ 1-18 and DOGL5.2. The latter has no heme-binding ability but is capable of interacting with AHG1 (Fig. 7b and Supplementary Fig. 7a). In contrast, the former retains heme-binding ability but abolishes the interaction with AHG1 (Fig. 6b and Supplementary Fig. 14d in the revised manuscript). In addition, we conclude that both binding of DOG1 to AHG1 and heme are essential for DOG1 function *in vivo* because neither YFP-DOG1 Δ 1-18ox nor YFP-DOGL5.2ox lines displayed ABA hypersensitive phenotypes in seed germination (Fig. 6a and Supplementary Fig. 8c,d and 12c).

5. The interpretation of DOG1 H245A and DOG1 H245A/H249A overexpression phenotypes is complicated by the lack of knowledge on the relationship between the overexpressed mutants and endogenous wild-type protein. In this regard, it also helps to know the oligomerization states of DOG1, DOGL3 and DOGL5.2.

Response: We thank the reviewer for the comment. We show that the N-terminal portion of DOG1 interacts with AHG1 and affects DOG1 function in plant (Fig. 6). Nakabayashi et al (2015) reported that the same region of DOG1 responsible for AHG1 interaction is a self-dimerization site. We show that H245A and H245A/H249A mutations in DOG1 result in

reduced heme binding activity and perturbed heme coordination *in vitro*, but did not abolish the interactions with AHG1 *in vivo* (Supplementary Fig. 14e in the revised manuscript). The oligomerization in DOG1 is clearly discussed in the manuscript and will be focus on our future research.

Further notes on the DOG1-heme interaction:

6. It looks like that DOG1 and DOGL3 are expressed with N-terminal His6 tags and DOGL5.2 is expressed with a C-terminal His6 tag. These tags are not removed during purification. The presence of the His6 tags can complicate the *in vitro* heme binding studies as they can also bind heme. Conversely, imidazole commonly used in His affinity purification can interact with free heme and favor dissociation of heme from the protein during the elution and any incubation before imidazole is removed. Usually it is better to use apo proteins without His tags in binding assays, and to avoid using imidazole during purification (for example employ ion exchange chromatography methods).

Response: We thank the reviewer for the comments. According to the suggestions, we performed the hemin titration experiments using the newly prepared native DOG1 without His tag, and obtained K_d of 59 nM, a 1.4-fold smaller value compared with that for the N-terminally His₆-tagged DOG1 (84 nM). We also newly prepared the N-terminally His₆-tagged DOGL5.2 and compared its absorption spectrum with that of the C-terminally His₆-tagged DOGL5.2, demonstrating that both spectra are exactly the same. To show these results clearly, we have replaced the data. Accordingly, we have revised Fig. 7c,d which show the data for the native DOG1 and moved the data for the His₆-tagged DOG1 to the newly prepared Supplementary Fig. 13. In addition, we have replaced the spectrum of the C-terminally His₆-tagged DOGL5.2 to that of the N-terminally His₆-tagged DOGL5.2 in the revised Fig. 7a,b.

7. Two methods were used to prepare hemin solutions when studying DOG1-heme interaction, namely dissolution in either 0.1 M NaOH and in DMSO. Note that hemin has a chloride group serving as a strong ligand. When dissolved in high concentrations of NaOH, chloride is replaced by OH⁻. This does not happen in DMSO. The two methods should be compared to be sure that no differences arise.

Response: We thank the reviewer for the comments. At least, the fully heme-bound DOG1s prepared by two different methods display essentially the same absorption spectra including the peak positions and their intensities. To make this point clear, we have added this statement in the Methods of the revised manuscript.

8. Fig. 7c,d, it would be helpful to the readers if the apoDOG1 protein concentration is explicitly given in the figure legend. By the way, line 582, “mM” perhaps should be “ μ M.” To understand how heme absorbs light after the DOG1 protein has been saturated, it would be interesting to generate a difference spectrum between 20 μ M and 12 μ M heme (the latter is a rough estimate by visual inspection). Also, relevant to the point above, the contribution of heme to extinction coefficient at 280 nm may be extrapolated from the titration.

Response: We are sorry for making such mistakes. We have mentioned the apo-protein concentrations of DOG1 and its mutants used for the hemin titration experiments in the legends for Fig. 7 and newly prepared Supplementary Fig. 13, and revised the line 582 of the

original manuscript. According to the suggestion, we have shown difference spectra generated by subtracting absorption spectra of hemin concentrations of 15, 16, 17, 18, 19 μM from that of 20 μM in Supplementary Fig. 13i-l. The obtained difference spectra differ depend on the values of the active hemin fraction (x) as well as K_d .

9. Fig. 7d, the extinction coefficient of DOG1-bound heme may be estimated from the slope. Visual estimate suggest a value of around 70 $\text{mM}^{-1}\text{cm}^{-1}$.

Response: We thank the reviewer for the suggestion. We have estimated the extinction coefficient of the DOG1-bound heme at 425 nm by nonlinear curve fitting as mentioned in the reply to the comment 2. The estimated values were 79 $\text{mM}^{-1}\text{cm}^{-1}$ for the native DOG1 and 84 $\text{mM}^{-1}\text{cm}^{-1}$ for the His₆-tagged DOG1. These values were used to estimate concentrations of the chromatographically purified proteins as mentioned in the reply to the comment 1.

10. 5 mM DTT is included in the protein and hemin solutions. The effects of DTT on the absorption spectra should be evaluated. First, DTT may serve as an exogenous co-axial ligand. Second, DTT at 5 mM concentration could reduce heme (although slowly). I suggest recording electronic absorption spectra in the absence of DTT, as well as preparing hemin solutions without DTT in the titration experiments.

Response: We thank the reviewer for the comments. However, DTT is required because DOG1 and its related proteins have tendency to oligomerize without DTT due to intermolecular disulfide bond formation. That is why we included 5 mM DTT in the protein and hemin solutions.

Minor comments on other parts of the manuscript:

11. There seems no strong need to define AHIPs since every one of them already has a name. This will reduce confusion.

Response: We thank the reviewer for this suggestion. We have removed it in the revised manuscript.

12. The subcellular localization studies rely on overexpression of fusions with fluorescence proteins. Both overexpression and fusion with a large tag protein could result in differences with the native proteins and potential misleading conclusions.

Response: We thank the reviewer for the comments. For overexpressing AHG1, we used two types of tags YFP and HA and showed that both YFP-AHG1ox and HA-AHG1ox lines exhibited ABA insensitive phenotypes, consistent with the data of our previously report using native AHG1ox lines (Nishimura et al 2007). For DOG1, we presented that DOG1 has the ability to bind AHG1 and heme. We show that H245A/H249A mutation in DOG1 results in reduced heme binding activity, but did not abolish the interactions with AHG1 (Supplementary Fig. 14e in the revised manuscript). In contrast, DOG1 Δ 1-18 retains heme-binding ability but abolishes the interaction with AHG1. YFP-DOG1ox lines have lower post-germination growth efficiency compared to YFP control plants (Fig. 4c), but YFP-DOG1^{H245AH249A}ox and YFP-DOG1 Δ 1-18OX lines show post-germination growth efficiency similarly to YFP controls (Fig. 6a and 8d), suggesting that YFP-DOG1 functions in plants. We understand the reviewer comments, but we have performed these experiments

carefully and believe our conclusions are supported by our present data.

13. Fig. 8d, is the value for YFP-DOG1 at 0.3 μ M ABA zero?

Response: Yes, it is zero.

References

Umezawa, T. *et al.* Type 2C protein phosphatases directly regulate abscisic acid-activated protein kinases in *Arabidopsis*. *Proc. Natl Acad. Sci. USA* **106**, 17588-17593 (2009).

Nakabayashi, K., Bartsch, M., Ding, J. & Soppe, W.J. Seed Dormancy in *Arabidopsis* Requires Self-Binding Ability of DOG1 Protein and the Presence of Multiple Isoforms Generated by Alternative Splicing. *PLoS Genet.* **11**, e1005737 (2015).

Nishimura, N. *et al.* *ABA-Hypersensitive Germination1* encodes a protein phosphatase 2C, an essential component of abscisic acid signaling in *Arabidopsis* seed. *Plant J.* **50**, 935-949 (2007).

Reviewers' comments:

Reviewer #5 (Remarks to the Author):

In this revision, the authors present new experiments, analyses, and interpretation. My major concerns #1 and #5-10 have been addressed. However, I still think that the K_d measurements are questionable, which in turn affect the subsequent functional analysis of the DOG1-heme interaction and dissection of the relationship between heme binding and AHG1-binding. I suggest a simple experiment that should allow quantitative comparison of the two mutants with WT.

Major issues:

#2, #3 & #4. The K_d measurements of DOG1 with heme remain a concern. I don't believe that the heme titration followed by non-linear curve fitting can provide true K_d values. Because the concentration of the proteins used in these experiments are around 10 μM , the method cannot be used to measure K_d values low than that. To convince the authors about this, I suggest that they redo the fitting with K_d values fixed to 1 nM, 10 nM, 100 nM, or even 1 μM . I predict that these values will result in fits as good as what's reported in the paper. If I am correct, the heme-titration experiments in the current manuscript do not provide evidence supporting that the heme affinities of DOG1 WT, H245A, and H245A/H249A mutants are different from each other. As such, the correlation heme binding affinity with DOG1 biological function and AHG1-binding becomes questionable. With that said, the bacterial overexpression and purification experiments do suggest that the heme affinity of the WT DOG1 is higher than those of the two mutants, but such experiments are not quantitative and the results can be variable. I suggest that the authors measure the kinetic dissociation rate as a proxy of heme affinity by incubating each heme-loaded DOG1 protein with large ($\sim 6x$) molar excess of apomyoglobin, which serves as a heme scavenger and is commercially available. John Olson's lab showed that the kinetic association rates of many heme proteins are similar.

#3. The absorption peaks of DOG1 mutant-heme complexes should be measured and labeled in Fig. 8. Apparently, the Soret peak remains pretty strong even when both His245 and His249 are mutated. This seems to argue against the conclusion about the heme iron ligation model. On the other hand, there seems to be a shift of the Soret peak by a few nm. Such a shift would be consistent with ligand switching. I think these should be delineated for the readers.

Minor comments:

a. Perhaps it is better to call "native DOG1" "untagged DOG1", since "native" is most often used to describe proteins from the natural sources.

b. Line 879, "x" represents a fraction of the active, likely monomeric, heme." It is pretty well known in the field that heme likes to stack with each other to form oligomers. Therefore, the fitted x values ranging from 0.81 to 1.0 cannot be monomeric.

Detailed response to reviewer:

We thank the reviewer for the comments. We have revised the manuscript and have attached the file including new data with detailed descriptions to reviewer comments. We have addressed the reviewer comments point-by-point as described in the following. Discussed changes are highlighted in the manuscript text.

Reviewer #5:

In this revision, the authors present new experiments, analyses, and interpretation. My major concerns #1 and #5-10 have been addressed. However, I still think that the K_d measurements are questionable, which in turn affect the subsequent functional analysis of the DOG1-heme interaction and dissection of the relationship between heme binding and AHG1-binding. I suggest a simple experiment that should allow quantitative comparison of the two mutants with WT.

Major issues:

Comment 1: #2, #3 & #4. The K_d measurements of DOG1 with heme remain a concern. I don't believe that the hemin titration followed by non-linear curve fitting can provide true K_d values. Because the concentration of the proteins used in these experiments are around 10 μM , the method cannot be used to measure K_d values low than that. To convince the authors about this, I suggest that they redo the fitting with K_d values fixed to 1 nM, 10 nM, 100 nM, or even 1 μM . I predict that these values will result in fits as good as what's reported in the paper. If I am correct, the heme-titration experiments in the current manuscript do not provide evidence supporting that the heme affinities of DOG1 WT, H245A, and H245A/H249A mutants are different from each other. As such, the correlation heme binding affinity with DOG1 biological function and AHG1-binding becomes questionable. With that said, the bacterial overexpression and purification experiments do suggest that the heme affinity of the WT DOG1 is higher than those of the two mutants, but such experiments are not quantitative and the results can be variable. I suggest that the authors measure the kinetic dissociation rate as a proxy of heme affinity by incubating each heme-loaded DOG1 protein with large (~6x) molar excess of apomyoglobin, which serves as a heme scavenger and is commercially available. John Olson's lab showed that the kinetic association rates of many heme proteins are similar.

Response: Although the reviewer has raised a question about the K_d values estimated by our nonlinear curve fitting of the experimental data, we are confident in our results. We believe that nonlinear curve fitting of our current experimental data could provide K_d values accurately by carefully examining three variable parameters, ϵ_{DH} , ϵ_{H} , and x . See an attachment file for assessment of the hemin titration curve fitting results. According to the suggestion by the reviewer, we repeated the curve fitting with the fixed K_d values in a range from 1 nM to 1 μM . However, the greater deviation of K_d from the best fit value is, the worse than the fitting result is. This is true for all the DOG1-related proteins. Therefore, we conclude that the heme-binding affinity is highest with wild-type DOG1, followed by H245A than H245AH249A mutants. This conclusion is further supported by the bound heme contents

in the recombinant wild-type DOG1 and its H245A and H245AH249A mutants expressed under the same condition II as clearly shown in Supplementary Fig. 14c. In addition, the heme-binding mode is different between these three proteins as suggested by their electronic absorption spectra (a revised Fig. 8 is now included where absorption peak positions of the DOG1 mutant-heme complexes are labeled). Both the differences in affinity and binding mode should affect biological activity of DOG1 and its mutants as demonstrated by our *in vivo* study.

Although the reviewer has recommended us to measure the kinetic dissociation rate, we do not think that such experiments are necessary because our K_d values estimated by our nonlinear curve fitting of the hemin titration experiments are quite accurate. Completion of the proposed experiments in the detail we strive for could take a long time. Since the kinetic dissociation rate should provide useful information regarding heme-binding mechanism, we have a plan to measure this value in our future research but feel it would unnecessarily further delay this manuscript.

Comment 2: #3. The absorption peaks of DOG1 mutant-heme complexes should be measured and labeled in Fig. 8. Apparently, the Soret peak remains pretty strong even when both His245 and His249 are mutated. This seems to argue against the conclusion about the heme iron ligation model. On the other hand, there seems to be a shift of the Soret peak by a few nm. Such a shift would be consistent with ligand switching. I think these should be delineated for the readers.

Response: We thank the reviewer for this comment. Peak positions of the H245A-bound heme shifted to 420 nm from 425 nm for the corresponding peak of the wild-type DOG1-bound heme. Peak positions of the H245AH249A-bound heme shifted further to 415 and 392 nm. As suggested by the reviewer, we have labeled the absorption peaks of DOG1 mutant-heme complexes in the revised Fig. 8.

Minor comments:

a. Perhaps it is better to call “native DOG1” “untagged DOG1”, since “native” is most often used to describe proteins from the natural sources.

Response: We thank the reviewer for this comment. As suggested by the reviewer, we have changed “native DOG1” to “untagged DOG1”.

b. Line 879, ““x” represents a fraction of the active, likely monomeric, hemin.” It is pretty well known in the field that hemin likes to stack with each other to form oligomers. Therefore, the fitted x values ranging from 0.81 to 1.0 cannot be monomeric.

Response: We thank the reviewer for this comment. According to the reviewer’s suggestion, we have changed “active, monomeric hemin” to “active hemin”.

Assessment of the hemin titration curve fitting results

We first performed linear regression analysis of the hemin titration experimental data for the untagged DOG1 to obtain approximate values of the extinction coefficients of the protein-bound heme, ϵ_{DH} , and the free hemin, ϵ_H , as well as “x” representing a fraction of the active hemin which can be incorporated into the protein (Fig. 1). The analyses of the lower and higher hemin concentration ranges from 0 to 5 μM and from 15 to 20 μM respectively provided approximate values of ϵ_{DH} and ϵ_H to be 68.2 and 29.3 $\text{mM}^{-1} \text{cm}^{-1}$, the latter is the upperlimit for ϵ_H . The two regression lines intersect at the hemin concentration of 11.36 μM (Fig. 1). Dividing the protein concentration, 9.4 μM , by this value of the intersection point, we could estimate a value of x of 0.827.

Then we performed nonlinear curve fitting of the entire experimental data with the following equations 1-3 with four variable parameters, ϵ_{DH} , ϵ_H , x and K_d . Results are summarized in Table 1.

$$A_{\text{obs}} = \epsilon_{DH}[\text{DH}] + \epsilon_H[\text{H}]_f \quad (1)$$

$$[\text{DH}] = ([\text{D}]_{\text{total}} + x[\text{H}]_{\text{total}} + K_d - \{([\text{D}]_{\text{total}} + x[\text{H}]_{\text{total}} + K_d)^2 - 4x[\text{H}]_{\text{total}}[\text{D}]_{\text{total}}\}^{1/2})/2 \quad (2)$$

$$[\text{H}]_f = [\text{H}]_{\text{total}} - [\text{DH}] \text{ and } [\text{D}]_{\text{total}} = [\text{DH}] + [\text{D}]_f \quad (3)$$

where [DH] represents the concentration of the heme-bound protein while $[\text{D}]_f$ and $[\text{H}]_f$ are concentrations of the free protein and free hemin, respectively.

First, we performed a two-parameter curve fitting with ϵ_H and x fixed respectively to 29.3 $\text{mM}^{-1} \text{cm}^{-1}$ and 0.827 (Fit-1), the values obtained by the linear regression analysis. Best fit provided $K_d = 33.3 \text{ nM}$ and $\epsilon_{DH} = 77.5 \text{ mM}^{-1} \text{cm}^{-1}$ (Fig. 2A and Table 1). A three-parameter curve fitting with only ϵ_H fixed to 29.3 $\text{mM}^{-1} \text{cm}^{-1}$ (Fit-2) provided nearly identical best fit with $K_d = 32.4 \text{ nM}$, $\epsilon_{DH} = 77.5 \text{ mM}^{-1} \text{cm}^{-1}$, and x = 0.826. These analyses may provide the lower limit of ϵ_{DH} as 77.5 $\text{mM}^{-1} \text{cm}^{-1}$. We also

performed a three-parameter curve fitting with x fixed to 0.827 and obtained the best fit with $K_d = 45.3$ nM, $\epsilon_{DH} = 78.0$ mM⁻¹cm⁻¹, and $\epsilon_H = 28.7$ mM⁻¹cm⁻¹ (Fit-3).

Next, we performed a three-parameter curve fitting with ϵ_{DH} fixed to 77.5 mM⁻¹cm⁻¹ (Fit-4) and obtained the best fit with $K_d = 29.7$ nM, $\epsilon_H = 29.2$ mM⁻¹cm⁻¹, and $x = 0.825$ (Fig. 3A). The same three-parameter curve fitting with ϵ_{DH} fixed to 78.0 mM⁻¹cm⁻¹ (Fit-5) provided the best fit with $K_d = 37.2$ nM, $\epsilon_H = 28.7$ mM⁻¹cm⁻¹, and $x = 0.821$ (Fig. 3B). It should be worth mentioning that the three-parameter curve fitting with ϵ_{DH} fixed to 77.5 (Fit-4) and 78.0 mM⁻¹cm⁻¹ (Fit-5) resulted in ϵ_H values larger than its upper limit, 29.3 mM⁻¹cm⁻¹, when used K_d values larger than 40 and 90 nM, respectively (Fig. 3C).

As suggested by the above curve fittings, three parameters, ϵ_{DH} , ϵ_H , and x , are coupled one from another. Therefore, we performed a four-parameter curve fitting with all three parameters treated as variable to determine actual values of these three simultaneously (Fit-6). The best fit was obtained with $K_d = 58.8$ nM, $\epsilon_{DH} = 79.4$ mM⁻¹cm⁻¹, $\epsilon_H = 27.4$ mM⁻¹cm⁻¹, and $x = 0.810$ (Table 1). Three-parameter fittings with either one of ϵ_{DH} , ϵ_H , x fixed to its best fit value (Fit-7 with $\epsilon_{DH} = 79.4$ mM⁻¹cm⁻¹, Fit-8 with $\epsilon_H = 27.4$ mM⁻¹cm⁻¹, and Fit-9 with $x = 0.810$) as well as two-parameter fittings with either two of ϵ_{DH} , ϵ_H , x fixed to their best fit values provided the exactly same results as the four-parameter fitting as shown in Table 1. These results suggest that our nonlinear curve fittings estimate the K_d value as well as the ϵ_{DH} , ϵ_H , and x values quite accurately.

According to the suggestion by the reviewer #5, we next have redone the four-parameter fitting (Fit-6) with the fixed K_d values in a range from 1 nM to 1 μ M. We also performed similar fittings for Fit-7, Fit-8 and Fit-9. The obtained best fit values of

ϵ_{DH} , ϵ_H , x as well as the $\chi^2 = (A_{\text{calc}} - A_{\text{obs}})^2$ value are plotted as a function of K_d (Fig. 4). The K_d dependence of ϵ_{DH} , ϵ_H , x , and χ^2 are quite different from one fitting to another although the four fittings provide exactly the same best fit with $K_d = 58.8$ nM, $\epsilon_{DH} = 79.4$ mM⁻¹cm⁻¹, $\epsilon_H = 27.4$ mM⁻¹cm⁻¹, and $x = 0.810$. For the four-parameter fitting Fit-6, the ϵ_H exceeds its upperlimit value of 29.3 mM⁻¹cm⁻¹ when the K_d value smaller than 11 nM is used (Fig. 4A), suggesting that the experimental data could not be fit accurately with K_d value smaller than 11 nM. For the three-parameter fitting Fit-7 with $\epsilon_{DH} = 79.4$ mM⁻¹cm⁻¹, the ϵ_H exceeds its upper limit value of 29.3 mM⁻¹cm⁻¹ when the K_d value larger than 252 nM is used (Fig. 4A), suggesting that the experimental data could not be fit accurately with K_d value larger than 252 nM. These results suggest that the actual K_d value must be in a range from 11 to 252 nM at the largest estimate. Similar to the K_d dependence of ϵ_{DH} , deviations of ϵ_H and x from their best fit values are getting significantly larger as the deviation of K_d from its best fit value of 58.8 nM increases (Fig. 4B and C). Furthermore, the goodness of fit, χ^2 , is also getting worse with a K_d value largely different from its best fit value (Fig. 4D). These results indicate that we are able to estimate the K_d value by nonlinear curve fitting of our hemin titration experiments with careful examination of the accuracy of three parameters, ϵ_{DH} , ϵ_H , and x .

Figure 5 shows the experimental data and theoretical absorption curves calculated with the best fit parameters of $\epsilon_{DH} = 79.4$ mM⁻¹cm⁻¹, $\epsilon_H = 27.4$ mM⁻¹cm⁻¹, and $x = 0.810$. The theoretical curve calculated with $K_d = 58.8$ nM fits to the experimental data quite nicely, of course. However, it is obvious that the theoretical curves calculated with $K_d = 1, 200, 500$ nM, and 1 μ M do not fit to the same experimental data.

Taken all the results mentioned above together, we reached the conclusion that

the K_d value for the untagged DOG1 is 58.8 nM which is associated with reasonably well defined ϵ_{DH} , ϵ_H and x values. Since similar results were obtained for the other DOG1-related proteins, we believe that we have correctly estimated K_d values for the heme-bindings of the wild-type DOG1 and its H245A and H245AH249A mutants and that they are different from each other.

Table 1. Parameters derived from nonlinear curve fitting^{a,b} the electronic absorption spectral data of hemin titration experiment for the untagged DOG1

	K_d	ϵ_{DH}	ϵ_H	x	χ^2
	(nM)	(mM ⁻¹ cm ⁻¹)	(mM ⁻¹ cm ⁻¹)		
Linear		68.2	29.3	0.827	
Fit-1	33.3	77.5	29.3	0.827	0.000304
	1	76.8	29.3	0.827	0.000663
	1000	87.1	29.3	0.827	0.00704
Fit-2	32.4	77.5	29.3	0.826	0.000303
	1	77.0	29.3	0.817	0.000593
	1000	84.4	29.3	0.931	0.00489
Fit-3	45.3	78.0	28.7	0.827	0.000285
	1	76.3	30.2	0.827	0.000580
	1000	94.1	21.7	0.827	0.00298
Fit-4	29.7	77.5	29.2	0.825	0.000299
	1	77.5	28.7	0.810	0.000663
	1000	77.5	34.3	1.088	0.00788
Fit-5	37.2	78.0	28.7	0.821	0.000267
	1	78.0	28.1	0.803	0.000790
	1000	78.0	33.8	1.073	0.00754
Fit-6	58.8	79.4	27.4	0.810	0.000239
	1	76.5	29.9	0.823	0.000573
	1000	127.2	2.6	0.638	0.00187
Fit-7	58.8	79.4	27.4	0.810	0.000239
	1	79.4	26.4	0.785	0.00130
	1000	79.4	32.7	1.038	0.00676
Fit-8	58.8	79.4	27.4	0.810	0.000239
	1	78.3	27.4	0.801	0.000924
	1000	86.9	27.4	0.895	0.00419
Fit-9	58.8	79.4	27.4	0.810	0.000239
	1	77.2	29.2	0.810	0.000633
	1000	95.9	20.5	0.810	0.00278

^a Best fit values were determined using following equations.

$$A_{\text{obs}} = \epsilon_{DH}[\text{DH}] + \epsilon_H[\text{H}]_f \quad (1)$$

$$[\text{DH}] = ([\text{D}]_{\text{total}} + x[\text{H}]_{\text{total}} + K_d - \{([\text{D}]_{\text{total}} + x[\text{H}]_{\text{total}} + K_d)^2 - 4x[\text{H}]_{\text{total}}[\text{D}]_{\text{total}}\}^{1/2})/2 \quad (2)$$

$$[\text{H}]_f = [\text{H}]_{\text{total}} - [\text{DH}] \text{ and } [\text{D}]_{\text{total}} = [\text{DH}] + [\text{D}]_f \quad (3)$$

^b Values in italic were fixed for the fitting.

Figure 1. Linear regression analysis of the electronic absorption spectral data of hemin titration experiment for the untagged DOG1.

Figure 2. Nonlinear curve fittings of the electronic absorption spectral data of hemin titration experiment for the untagged DOG1. (A) A two-parameter fitting with ϵ_H and x fixed respectively to $29.3 \text{ mM}^{-1}\text{cm}^{-1}$ and 0.827 , the values obtained by linear regression analysis. (B) A three-parameter fitting with ϵ_H fixed to $29.3 \text{ mM}^{-1}\text{cm}^{-1}$. (C) A three-parameter fitting with x fixed 0.827 .

Figure 3. Nonlinear curve fittings of the electronic absorption spectral data of hemin titration experiment for the untaged DOG1. (A) A three-parameter fitting with ϵ_{DH} fixed to 77.5 mM $^{-1}$ cm $^{-1}$, the lower limit value obtained by Fit-1 and Fit-2. **(B)** A three-parameter fitting with ϵ_{DH} fixed to 78.0 mM $^{-1}$ cm $^{-1}$, the best fit value obtained by Fit-3. **(C)** Relationship between ϵ_H and K_d obtained by Fit-4 and Fit-5. **(D)** Relationship between x and K_d obtained by Fit-4 and Fit-5.

Figure 4. K_d dependence of (A) ϵ_H , (B) ϵ_{DH} , (C) x , and (D) χ^2 obtained by Fit-6, Fit-7, Fit-8, and Fit-9.

Figure 5. The electronic absorption data of hemin titration experiment for the untagged DOG1 (●) and theoretical curves calculated with the best fit values of $\epsilon_{DH} = 79.4 \text{ mM}^{-1}\text{cm}^{-1}$, $\epsilon_H = 27.4 \text{ mM}^{-1}\text{cm}^{-1}$, and $x = 0.81$: $K_d = 1$ (light blue line), 58.8 (black line), 200 (orange line), 500 nM (light green line), and 1 μM (red line).

REVIEWERS' COMMENTS:

Reviewer #5 (Remarks to the Author):

The authors have addressed all my concerns. From their careful fitting exercise, I am convinced that the heme titration data do indicate differential heme affinities of the wild-type and mutant DOG1 proteins, thereby providing strong support to the conclusion. Therefore, I recommend publication of this excellent manuscript in Nature Communication.